# Freshwater gobies 30 million years ago: New insights into character evolution and phylogenetic relationships of †Pirskeniidae (Gobioidei, Teleostei)

**Bettina Reichenbacher**[1,2]*, **Tomáš Přikryl**[3], **Alexander F. Cerwenka**[4], **Philippe Keith**[5], **Christoph Gierl**[1], **Martin Dohrmann**[6]

**1** Department of Earth and Environmental Sciences, Ludwig-Maximilians-Universität München, Munich, Germany, **2** GeoBio-Center LMU, Ludwig-Maximilians-Universität München, Munich, Germany, **3** Institute of Geology of the Czech Academy of Sciences, Praha, Czech Republic, **4** Section Evertebrata varia, SNSB-Bavarian State Collection of Zoology, Munich, Germany, **5** Laboratoire de Biologie des organismes et écosystèmes aquatiques (BOREA), Muséum national d'Histoire naturelle, Centre National de la Recherche Scientifique, Institut de Recherche pour le Développement, Sorbonne Université, Paris, France, **6** SNSB-Bavarian State Collection of Palaeontology and Geology, Munich, Germany

* b.reichenbacher@lrz.uni-muenchen.de

**Data Availability Statement:** The new CT slice data and 3D files of *Rhyacichthys guilberti* (2 specimens), *Protogobius attiti* (1 specimen),

## Abstract

The modern Gobioidei (Teleostei) comprise eight families, but the extinct †Pirskeniidae from the lower Oligocene of the Czech Republic indicate that further families may have existed in the past. However, the validity of the †Pirskeniidae has been questioned and its single genus †*Pirskenius* has been assigned to the extant family Eleotridae in previous works. The objective of this study is to clarify the status of the †Pirskeniidae. Whether or not the †Pirskeniidae should be synonymised with the Eleotridae is also interesting from a bio-geographical point of view as Eleotridae is not present in Europe or the Mediterranean Sea today. We present new specimens and re-examine the material on which the two known species of †*Pirskenius* are based (†*P. diatomaceus* Obrhelová, 1961; †*P. radoni* Přikryl, 2014). To provide a context for phylogenetically informative characters related to the palatine and the branchiostegal rays, three early-branching gobioids (*Rhyacichthys*, *Protogobius*, *Perccottus*), an eleotrid (*Eleotris*) and a gobiid (*Gobius*) were subjected to micro-CT analysis. The new data justify revalidation of the family †Pirskeniidae, and a revised diagnosis is presented for both †*Pirskenius* and †Pirskeniidae. Moreover, we provide for the first time an attempt to relate a fossil gobioid to extant taxa based on phylogenetic analysis. The results indicate a sister-group relationship of †Pirskeniidae to the Thalasseleotrididae + Gobiidae + Oxudercidae clade. Considering the fossil record, the arrival of gobioids in freshwater habitats in the early Oligocene apparently had generated new lineages that finally were not successful and became extinct shortly after they had diverged. There is currently no evidence that the Eleotridae was present in the European ichthyofauna in the past.

*Eleotris pisonis* (2 specimens) and *Gobius incognitus* (1 specimen) are available from the MorphoSource repository (Project P1063, www. morphosource.org/Detail/ProjectDetail/Show/ project_id/1063).

**Funding:** We acknowledge funding for this project from the Deutsche Forschungsgemeinschaft to B. R. (RE- 1113/20-1). T.P.'s research was institutionally supported by the Czech Academy of the Sciences, Institute of Geology (RVO67985831). The funders had no role in study design, data collection and analysis, decision to publish, or preparation of the manuscript.

**Competing interests:** The authors have declared that no competing interests exist.

## Introduction

Living gobioids are distributed worldwide and constitute one of the most species-rich vertebrate suborders, with approximately 2,200 species belonging to > 270 genera [1]. They are small, mostly benthic fishes that form a significant faunal component of reefs and other shallow marine ecosystems, and are also abundant in brackish and freshwater habitats [2]. Among the extant Gobioidei, eight families have been recognised based on morphological characteristics and molecular phylogenetics, i.e. the Rhyacichthyidae, Odontobutidae, Milyeringidae, Eleotridae, Butidae, Thalasseleotrididae, Gobiidae, and Oxudercidae; Rhyacichthyidae and Odontobutidae are sister to all other gobioid families [3–7]. Gobiidae and Oxudercidae represent the most derived clade (Fig 1). These two share three derived characters that are in principle recognisable in fossils, namely the possession of five branchiostegal rays (vs. six in the other groups), a 'T-shaped' (vs. an 'L-shaped') palatine bone, and united pelvic fins [4, 8–10].

Although the first appearance of a lineage in the fossil record is always younger than the primary divergence event, the fossil record can provide crucial data on the lineage's distribution and diversification in the past. The oldest skeleton-based gobioid species from a marine environment is †*Carlomonnius quasigobius* Bannikov and Carnevale, 2016. It has been described based on a single well preserved specimen of very small size (13 mm standard length) from the lower Eocene (upper Ypresian, ca. 50 Ma) of Monte Bolca; its family status was left as *incertae sedis* [13]. The oldest gobioid species known from clearly freshwater environments are based on skeletons dated to the early Oligocene [14–17]. They are represented by four species, †*Pirskenius diatomaceus* Obrhelová, 1961, †*P. radoni* Přikryl, 2014, '*Gobius*' *gracilis* Laube, 1901 (which probably represents an extinct genus) and †*Lepidocottus papyraceus* (Agassiz, 1832). While †*L. papyraceus* is a member of the Butidae [16, 18], the family relationships of the others are either unknown

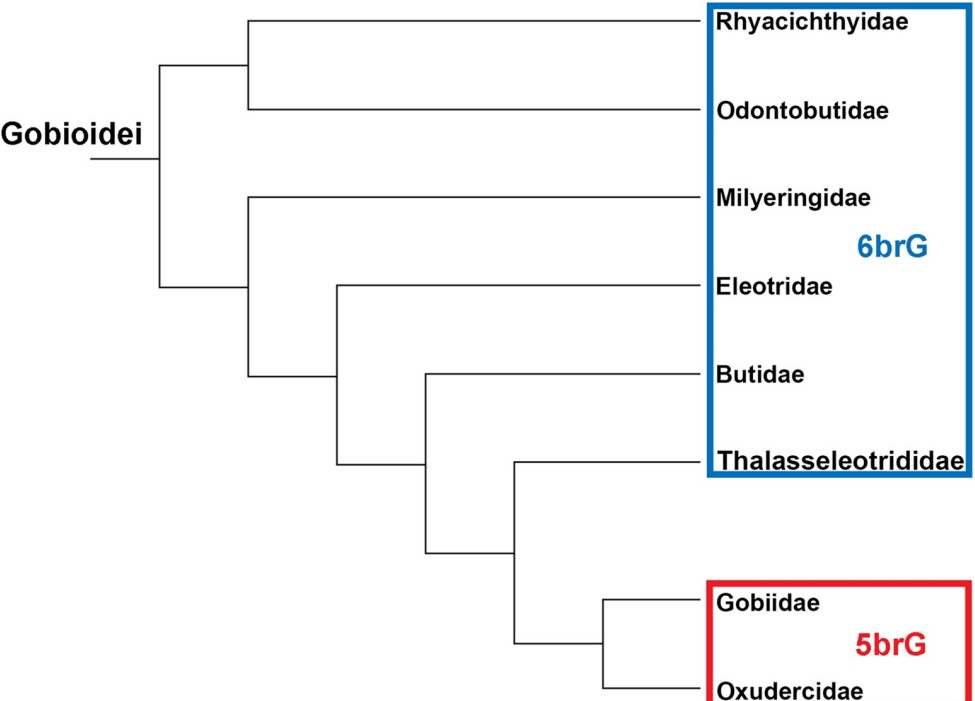

**Fig 1. Phylogeny of the Gobioidei according to Betancur et al. [11], Nelson et al. [1], and Thacker [6].**
6brG = gobioid families with six branchiostegal rays, 5brG = gobioid families with five branchiostegal rays. Modified after Gierl and Reichenbacher [12].

('*Gobius' gracilis*) or disputed (†*Pirskenius*). The original description of †*Pirskenius* by Obrhelová [15] considered it to represent a new, extinct family, †Pirskeniidae Obrhelová, 1961. However, it was later suggested that †Pirskeniidae cannot be sustained and that its single known genus †*Pirskenius* should be assigned to the Eleotridae [17, 19]. In that case, †*Pirskenius* would represent the oldest skeleton-based species of the family Eleotridae. In addition, it would indicate that eleotrids had become established in European inland waters by the early Oligocene.

This study set out to resolve the status of the †Pirskeniidae and thus contribute to a better understanding of the ancient diversity and biogeography of gobioid families. To achieve this, previously available †*Pirskenius* material and new finds were investigated in the light of a comparative micro-CT study of the branchiostegal rays and the palatine in three early-branching gobioids (*Rhyacichthys*, *Protogobius*, *Perccottus*), an eleotrid (*Eleotris*) and a gobiid (*Gobius*). For further comparisons of †*Pirskenius* and phylogenetic analysis of †Pirskeniidae we used morphological data from previous works [4, 9, 20–29].

## Geological setting

The type locality of †*Pirskenius diatomaceus* Obrhelová, 1961 is Knížecí, in the north of the Czech Republic, and the type locality of †*P. radoni* is Byňov, which lies about 60 km SW of Knížecí (Fig 2). Both sites are located in the volcanoclastic complex of the České Středohoří Mountains. During the Oligocene, this region experienced significant tectonic and volcanic activity, which also gave rise to the emergence of several freshwater lakes, as indicated by the fossil finds at Knížecí, Byňov and several other sites (Fig 2) [14, 17, 19]. Knížecí is a mine dump on the northern slope of the Hrazený hill (whose former name, 'Pirskensberg', accounts for the designation of the genus), east of the village of Knížecí, near Šluknov [15, 30]. The ancient outcrop was an approximately 5-m-thick succession of diatomites and sandy-to-coaly clays, overlain by tephritic rocks (Kopecký in [31]). The outcrop itself must have been re-filled or destroyed shortly after its initial description, because all fossils were collected from diatomites associated with the mine dump (Zlatko Kvaček, pers. comm., [15]). The fish fossils from Knížecí were made up solely of the gobioid †*P. diatomaceus* and the cyprinid †*Protothymallus elongatus* (Gorjanović-Kramberger, 1885) [14, 15, 19, 32]. In addition, well-preserved plants have been reported [31, 33]. Obrhelová [15] assumed †*P. diatomaceus* to be an Oligo-Miocene species because, at the time of her study, the age of the Knížecí deposit was not clear. Bellon et al. [30] provided an $^{40}$K-$^{40}$Ar age of 29.5±1.5 MYA (= early Oligocene) based on 'one surface sample taken at about 510 m in height'. As the original outcrop no longer existed at that time, this sample was probably collected from the mine dump.

Byňov, the type locality of †*P. radoni*, is also a mine dump, situated to the east of the village of Byňov [17]. Apart from four specimens of †*P. radoni*, several plant fossils have been described [34]. Their composition is similar to that of the floral assemblages from Seifhennersdorf and Kundratice (see Kvaček in [17]), and these sites are early Oligocene in age [14]. Accordingly, both Byňov and Knížecí can now be attributed to the early Oligocene.

**Institutional abbreviations.** MNHN, Muséum national d'Histoire naturelle, Paris, France; NMP, National Museum, Prague, Czech Republic; SNSB-ZSM, Zoological State Collection, Munich, Germany; UWFC, University of Washington, Burke Museum of Natural History and Culture, Seattle, Washington, USA.

## Materials and methods

### Ethic statement

Collection of *Rhyacichthys guilberti* Dingerkus and Séret, 1992 (MNHN 2019–0113) was granted by the Vanuatu Environment Unit (Permit Numbers ENV326/001/1/07/DK and

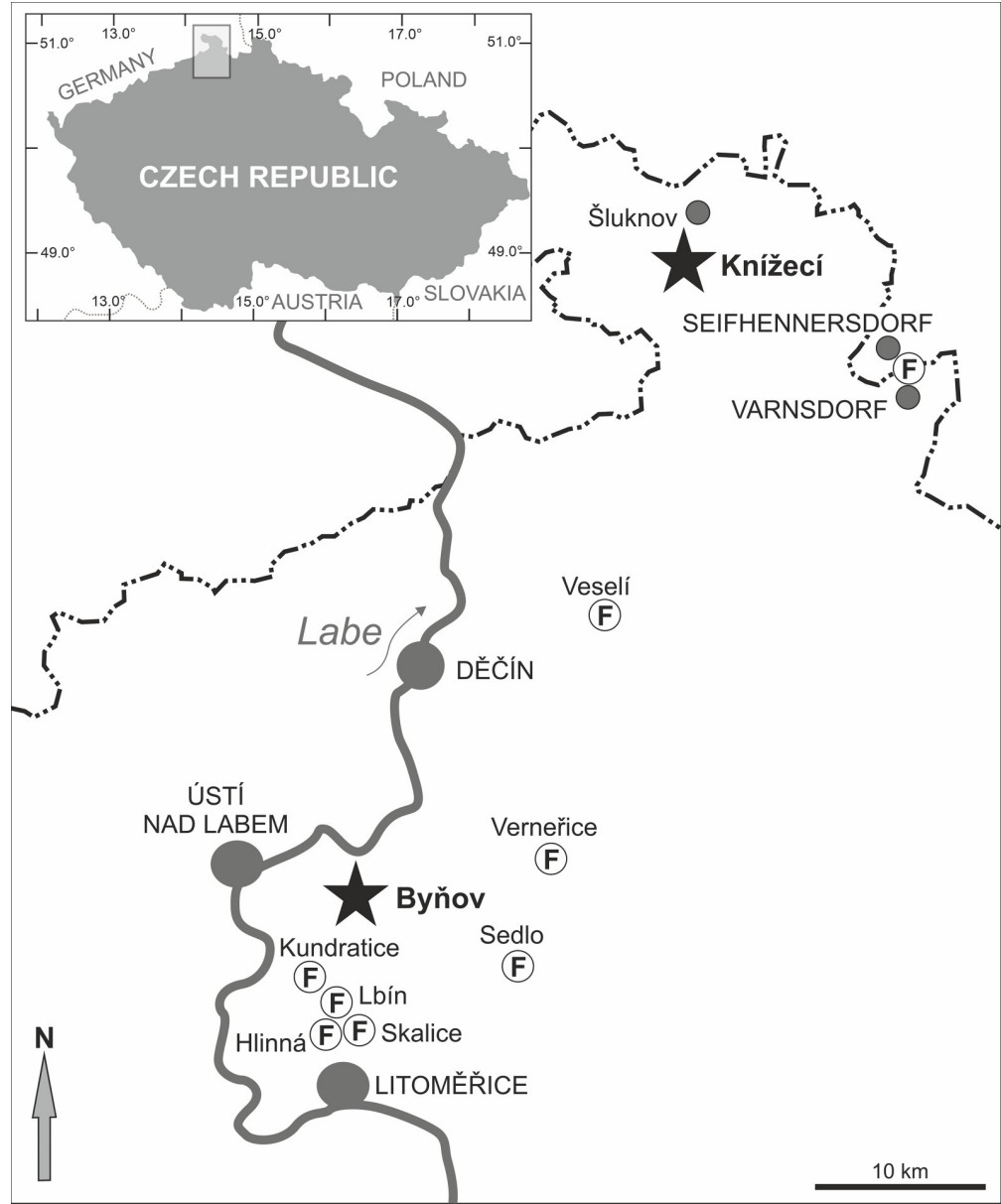

**Fig 2. Map of the area with the type localities (stars) of †*Pirkenius diatomaceus* and †*P. radoni* and other fossiliferous sites (F).** All sites are located in the České Středohoří Mountains (northern Czech Republic). Redrawn from Gaudant [19] under a CC BY license, with permission from Naturhistorisches Museum Wien, original copyright 2009.

ENV326/001/1/08/D). Collection of *Protogobius attiti* Watson and Pöllabauer, 1998 (MNHN 2019–0112) was permitted by the New Caledonian South Province (Permit Number 1224-08/PS).

## Fossil specimens

Three well preserved articulated fossil skeletons of †*Pirskenius* (collection numbers NMP Pv 11669, 11671, 11672) were newly available from Knížecí. They had been collected by Ervín Knobloch in 1958 and 1961, but were only recently discovered in the 'Knobloch collection' of

the Czech Geological Survey (see Kvaček et al. [33]); now they are stored in the National Museum in Prague. Further fossil specimens comprised 44 skeletons of the original material of †*P. diatomaceus* described by Obrhelová (1961) (collection numbers NMP PC 2770–PC 2822) and the four known specimens of †*P. radoni* described by Přikryl [17] (collection numbers MT PA1480–1484).

The definition of morphometric characters follows Liu et al. [35]. Morphometric measurements of †*P. diatomaceus* were taken with ImageJ v1.51a 64-bit [36] based on the digital images. Measurements of †*P. radoni* were adopted from Přikryl [17]. Measurements were standardised based on the standard length of the fossil; only in the case of the eye diameter standardization was based on the head length. Osteological and meristic characters were examined with a stereomicroscope (Leica MZ6, Leica M165 FC, Olympus SZX 12) or a Keyence digital microscope, each equipped with a digital camera (Canon EOS 1000D, Leica DFC 450, Olympus DP72, Keyence VH-Z20 UR). Topographic terms used to designate bones refer to their natural anatomical position, even if the bone was displaced in the fossil specimen. Counts of vertebrae include the terminal centrum, counts of rays in the second dorsal fin and anal fin include every discernible ray.

## Comparative material

Specimens of representative extant species of the families Rhyacichthyidae, Odontobutidae, Eleotridae and Gobiidae were used for comparative studies of the branchiostegal rays and the shape of the palatine head: (1) *Rhyacichthys guilberti* Dingerkus and Séret, 1992, Rhyacichthyidae (two specimens, Malekula island, Vanuatu, Oceania, MNHN 2019–0113); (2) *Protogobius attiti* Watson and Pöllabauer, 1998, Rhyacichthyidae (one specimen, New Caledonia, MNHN 2019–0112); (3) *Perccottus glenii* Dybowski, 1877, Odontobutidae (NW Sakhalin island, Russia, UWFC 44788); (4) *Eleotris pisonis* (Gmelin, 1789), Eleotridae, type species of *Eleotris* Bloch and Schneider, 1801 (two specimens, ZSM 9393 from Curaçao Island, Caribbean Sea, ZSM 41704 from Rio Cerere, Costa Rica); (5) *Gobius incognitus* Kovačić and Šanda, 2016, Gobiidae (one specimen, Pelješac peninsula, Croatia, NMP 6V 146150).

To examine their branchiostegal rays and the palatine, scans were taken for each species (except for *P. glenii*, see below) with a phoenix nanotom μ-CT-scanner equipped with a cone-beam scanner (GE Sensing and Inspection Technologies GmbH, Wunstorf, Germany). Fish specimens were mounted in a plastic vessel and stabilised with paper wipes. A small amount of 75% ethanol was added to the vessel to avoid drying. Specimens were scanned over 360˚ with three averaged images per rotation position, using a tungsten ('standard') target. Further details of each scan are given in Table 1. The μ-CT scans of *P. glenii* were available from the MorphoSource database (www.morphosource.org/; project 'Scan all Fishes', MorphoSource Identifier: S30430). The μ-CT data sets of the newly scanned species have also been uploaded to the MorphoSource database [37] (see Table 1 for details).

Subsequently, all datasets were reconstructed separately with the software phoenix datos|x 2 (GE Sensoring and Inspection Technologies GmbH, Germany), cropped and converted to 8bit using VGStudio MAX 2.2 (Volume Graphics, Heidelberg, Germany). Co-registering and merging of the multiscans and visualization were done manually using Amira 6.4 software (FEI Visualization Sciences Group, Burlington, MA, USA).

## Phylogenetic methods

**Character mapping on a molecular phylogeny.** Derived states of morphological characters according to Hoese and Gill [4], Gill and Mooi [9] and the results of the present study (Table 2) were mapped on a recently published molecular phylogeny of the Gobioidei

**Table 1. Technical details of the micro-CTs of extant gobioids.**

| | Voltage [kV] | Current [µA] | Resolution (x/y/z-axis) [µm] | Number of scans and projections per scan | MorphoSource Identifier: |
|---|---|---|---|---|---|
| *Rhyacichthys guilberti* MNHN 2019-0113-2 (small specimen) | 100 | 140 | 9.3 | 2*1440 | S32390 |
| *Rhyacichthys guilberti* MNHN 2019-0113-1 | 180 | 125 | 19.6 | 3*1600 | S32389 |
| *Protogobius attiti* MNHN 2019–0112 | 100 | 70 | 13.3 | 1440 | S32392 |
| *Perccottus glenii* UWFC 44788 | 65 | 123 | 13.8 | 800 | S30430 |
| *Eleotris pisonis* ZSM 9393, 41704 | 100 | 100 | 4.1 | 1*1200 | S32337, S32375 |
| *Gobius incognitus* NMP 6V 146150 | 110 | 50 | 16.1 | 3*1440 | S32391 |

The µ-CT scans of *R. guilberti* (2 specimens), *P. attiti* (1 specimen), *E. pisonis* (2 specimens) and *G. incognitus* (1 specimen) are available from the MorphoSource database [37]. The data of *P. glenii* derives from the MorphoSource database project oVert: UW—CT Scan all Fishes.

provided by Thacker et al. [7]. Plausible positions of the †Pirskeniidae (see Systematic Palaeontology section for justification) were determined as suggested by the character state distributions.

**Maximum parsimony analysis.** A phylogenetic analysis based on maximum parsimony was conducted using the taxon set and morphological data presented by Hoese and Gill [4]; some further morphological data were added from Hoese [26], Hoese and Larson [23], Gill and Mooi [9] and the outcome of this study (see Table 2 and Supporting Information S1 and S2 Files). The taxon set of Hoese and Gill [4] included the families Rhyacichthyidae, Odontobutidae, Butidae (as Butinae), Eleotridae (as Eleotridinae) and Gobiidae/Oxudercidae (as Gobiinae) (see Hoese and Gill [4]: Tables 1 and 2 and 'Osteological Material Examined' for included species; no specific outgroup had been defined). We adopted this taxon set, with the following modifications: the families Thalasseleotrididae and Milyeringidae (subsumed by Hoese and Gill [4] with the 'Eleotridinae' and 'Butinae', respectively) were added, and a generalized percomorph was included as outgroup. †Pirskeniidae (see Systematic Palaeontology section for justification) was incorporated into the data matrix by inserting character states for nine characters (for details see Table 2).

The entire list of characters comprises 20 phylogenetically informative characters, which are specified in Table 2. The original matrix as presented in Hoese and Gill [4] contained 16 osteological characters. Three of those characters were discarded here (preopercular mandibular canal; infraorbital canal; interhyal position) as their phylogenetic information solely concerned the Rhyacichthyidae and was the same as the character 'penultimate branchiostegal ray position', which–in contrast to the three discarded characters–can be easily recognized in fossils. In addition, the binary character 'branchiostegal ray number' of Hoese and Gill [4] was transformed into a new multistate character termed 'serial number of expanded last branchiostegal ray' (Table 2, character 12) because it is not simply the number of branchiostegal rays that is important, but also the fact that the last branchiostegal ray is expanded (see [38]). The other character states were the same as used by Hoese and Gill [4], except for characters 1 (adductor mandibulae tendon attachment), 3 (posterior extent of procurrent caudal cartilages) and 8 (penultimate branchiostegal ray position), which were modified from binary to multistate characters taking into account the osteological details described (but not coded) by Hoese and Gill [4] and Hoese and Larson [23]. To this list of characters, the palatine shape with a slightly refined definition of its character states (Table 2, character 14), and the synapomorphies recognized by Gill and Mooi [9] for the Thalasseleotrididae + Gobiidae/Oxudercidae and Gobiidae/Oxudercidae, respectively, were added (Table 2, characters 15–20). Furthermore, our matrix takes into account previously reported intra-family variation, namely with

**Table 2. Morphological characters for classification of Gobioidei.**

| Description of characters and states | Outgroup | Rhyacichthyidae | Odontobutidae | Milyeringidae | Eleotridae | Butidae | Thalasseleotrididae | Gobiidae | Oxudercidae | †Pirskeniidae |
|---|---|---|---|---|---|---|---|---|---|---|
| (1) *adductor mandibulae tendon attachment* (Hoese and Gill 1993: char. 1, modified from binary to multistate character; character state for Thalasseleotrididae according to Hoese and Larson 1987): 0 = posterior to or on maxilla head; 1 = on posterodorsolateral margin of maxilla; 2 = to short process extending from maxilla just posterior to maxilla head, in front of maxillo–dentary ligament; 3 = medially halfway down directly at maxilla shaft, posterior to maxillo–dentary ligament; 4 = various attachments | 0 | 1 | 2 | 2 | 3 | 2 | 2 | 4 | 4 | ? |
| (2) *anterior extent of procurrent cartilages* (Hoese and Gill 1993: char. 2): 0 = small; 1 = expanded | 0 | 0 | 0 | 1 | 1 | 1 | 1 | 1 | 1 | ? |
| (3) *posterior extent of procurrent caudal cartilages* (Hoese and Gill 1993: char. 3, modified from binary to multistate character): 0 = relatively short, not extending over epural; 1 = elongated, extending over anterior epural; 2 = elongated, covering full epural | 0 | 0 | 0 | 0 | 1&2 | 0 | 1 | 0&2 | 0&2 | ? |
| (4) scapula/radial position (Hoese and Gill 1993: char. 4): 0 = radial separate from cleithrum, large ossified scapula; 1 = radial adjoins cleithrum, scapula unossified or reduced | 0 | 0 | 0 | 1 | 1 | 1 | 1 | 1 | 1 | 1 |
| (5) middle radial of first pterygiophore of second dorsal fin (Hoese and Gill 1993: char. 5): 0 = present; 1 = fused with proximal radial (or lost) | 0 | 0 | 0 | 1 | 1 | 1 | 1 | 1 | 1 | ? |
| (6) bony preopercle canal support (Hoese and Gill 1993: char. 6): 0 = usually extends full length of preopercle; 1 = confined to vertical portion of preopercle or absent | 0 | 0 | 0 | 0 | 1 | 0&1 | 1 | 1 | 1 | 0 |
| (7) ctenoid scales with transforming ctenii (Hoese and Gill 1993: char. 7): 0 = present; 1 = absent | 1 | 0 | 0 | 1 | 1 | 1 | 1 | 1 | 1 | 1 |
| (8) penultimate branchiostegal ray position (Hoese and Gill 1993: char. 8; this study, modified from binary to multistate character): 0 = at posterior ceratohyal; 1 = at gap or just before gap to posterior ceratohyal; 2 = at anterior ceratohyal and clearly anterior to gap to posterior ceratohyal | 0 | 0&1 | 1 | 2 | 2 | 2 | 2 | 2 | 2 | 1 |

(*Continued*)

**Table 2.** (Continued)

| Description of characters and states | Outgroup | Rhyacichthyidae | Odontobutidae | Milyeringidae | Eleotridae | Butidae | Thalasseleotrididae | Gobiidae | Oxudercidae | †Pirskeniidae |
|---|---|---|---|---|---|---|---|---|---|---|
| (9) lateral line (Hoese and Gill 1993: char. 9): 0 = present; 1 = absent (state '0' for Odontobutidae refers to lateral line of *Terateleotris*; see Shibukawa et al. 2001) | 0 | 0 | 0&1 | 1 | 1 | 1 | 1 | 1 | 1 | 1 |
| (10) ventral shelf on urohyal (Hoese and Gill 1993: char. 12; Gill and Mooi 2012): 0 = present; 1 = absent | 0 | 0 | 0 | 0 | 0 | 0 | 1 | 1 | 1 | ? |
| (11) interneural gap between last pterygiophore of first dorsal fin and first pterygiophore of second dorsal fin (Hoese and Gill 1993: char. 13): 0 = absent; 1 = present (squared brackets refer to occasional occurrence of state '1' in the Eleotridae and Butidae; see Birdsong et al. 1988) | 0 | 0 | 0 | 0 | 0[&1] | 0[&1] | 1 | 1 | 1 | 1 |
| (12) serial number of expanded last branchiostegal ray (Hoese and Gill 1993: char. 15; Gill and Mooi 2012; this study; modified from binary to multistate character): 0 = absent; 1 = sixth ray; 2 = seventh ray; 3 = fifth ray | 0 | 1 | 1 | 1 | 1 | 1 | 1 | 3 | 3 | 2 |
| (13) Pelvic fins (Akihito 2000; Hoese and Gill 1993: char. 16): 0 = separated; 1 = united | 0 | 0 | 0 | 0 | 0 | 0 | 0 | 1 | 1 | 0 |
| (14) palatine shape (Regan 1911; Hoese 1984; Nelson 1986; Hoese and Larson 1987; this study): 0 = elongate; 1 = elevated maxillary process, short triangular ethmoid process; 2 = more or less L-shaped, no clear ethmoid process; 3 = not exactly L-shaped, short, slender ethmoid process; 4 = T-shaped, ethmoid process well developed and as long as maxillary process | 0 | 1&2 | 2 | 2 | 2 | 2 | 3 | 4 | 4 | 3 |
| (15) Interhyal with cup-shaped lateral structure for articulation with preopercle (Gill and Mooi 2012): 0 = not developed; 1 = present | 0 | 0 | 0 | 0 | 0 | 0 | 1 | 1 | 1 | ? |
| (16) laterally-directed posterior process on the posterior ceratohyal supporting the interhyal (Gill and Mooi 2012): 0 = not developed; 1 = present | 0 | 0 | 0 | 0 | 0 | 0 | 1 | 1 | 1 | ? |
| (17) *Pharyngobranchial 4 absent, epibranchial 4 directly articulating with pharyngobranchial 3* (Gill and Mooi 2012): 0 = not developed; 1 = present | 0 | 0 | 0 | 0 | 0 | 0 | 1 | 1 | 1 | ? |

(Continued)

Table 2. (Continued)

| Description of characters and states | Outgroup | Rhyacichthyidae | Odontobutidae | Milyeringidae | Eleotridae | Butidae | Thalasseleotrididae | Gobiidae | Oxudercidae | †Pirskeniidae |
|---|---|---|---|---|---|---|---|---|---|---|
| (18) dorsal postcleithrum (Hoese 1984; Gill and Mooi 2012): 0 = present, 1 = absent (squared brackets refer to occasional occurrence of state '1' in the Eleotridae; see Hoese 1984) | 0 | 0 | 0 | 0 | 0[&1] | 0 | 1 | 1 | 1 | 1 |
| (19) *expanded and medially-placed ventral process on ceratobranchial 5* (Gill and Mooi 2012): 0 = not developed; 1 = present | 0 | 0 | 0 | 0 | 0 | 0 | 0 | 1 | 1 | ? |
| (20) *dorsal hemitrich of pelvic-fin rays with complex proximal head* (Gill and Mooi 2012): 0 = not developed; 1 = present | 0 | 0 | 0 | 0 | 0 | 0 | 0 | 1 | 1 | ? |

Characters that refer to soft tissue or delicate bony structures and thus are unlikely to be preserved in a fossil are indicated in italics.

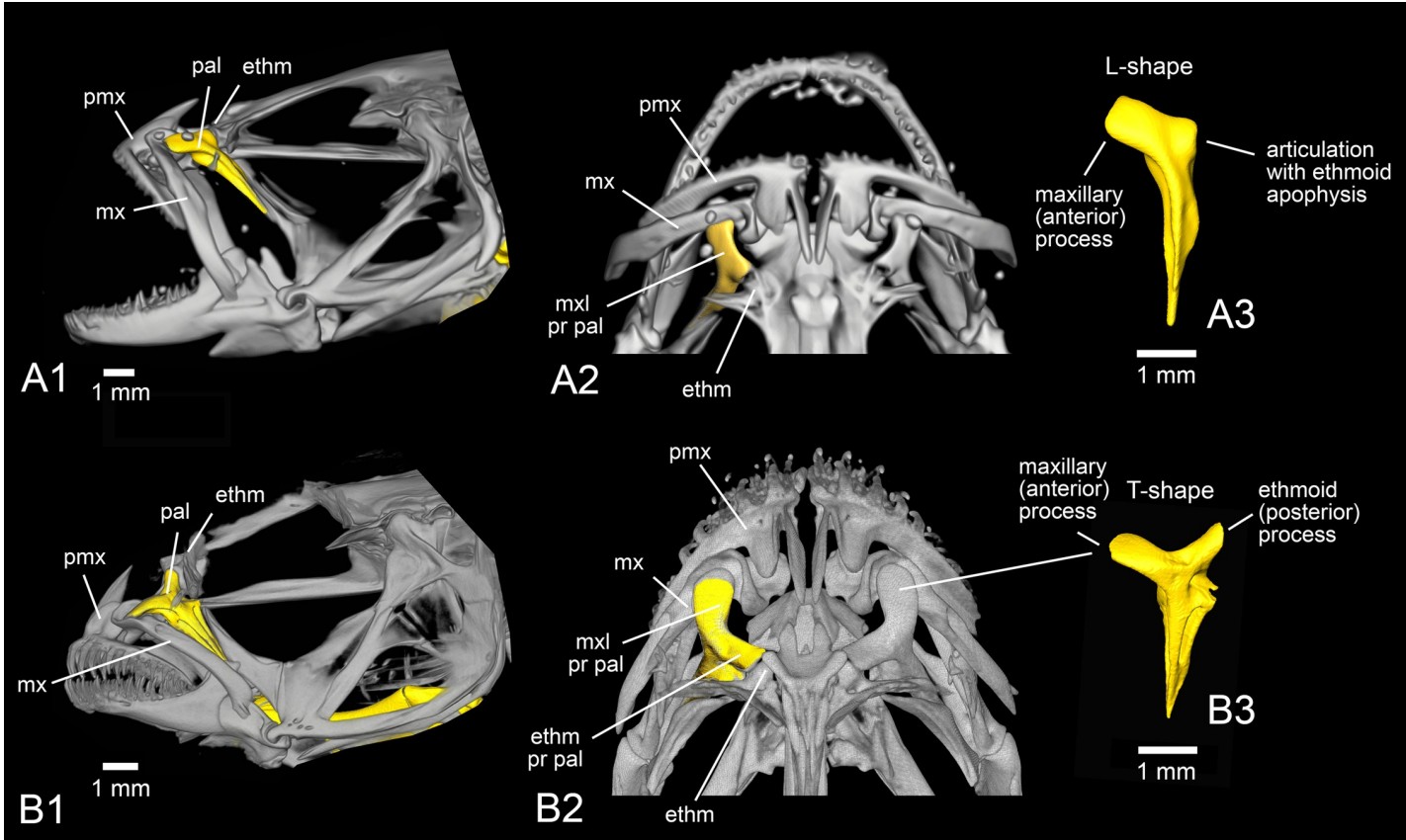

**Fig 3. Anatomical position and details of the palatine (yellow) of extant gobioid species.** (A) *Eleotris pisonis* (ZSM 9393), skull in lateral (A1) and dorsal (A2) views, left palatine (A3) in lateral view. (B) *Gobius incognitus* (NMP6V 146150), skull in lateral (B1) and dorsal (B2) views, left palatine (B3) in lateral view. All images based on μ-CT scanning. Abbreviations: ethm, ethmoid; ethm pr pal, ethmoid process of palatine; mx, maxilla; mxl pr pal, maxillary process of palatine; pal, palatine; pmx, premaxilla.

respect to the lateral line (character 9) of the Odontobutidae [28], the interneural gap (character 11) of the Eleotridae and Butidae [21], the palatine shape (character 14) of the Rhyacichthyidae (this study), and the dorsal postcleithrum of the Eleotridae [26].

The newly compiled matrix was edited in Mesquite 3.61 [39]. Phylogenetic analysis was performed under maximum parsimony in PAUP* 4.0a (build 167) [40]. Characters were treated as unordered and equally weighted and clade support was assessed using standard bootstrapping [41] with 1000 pseudo-replicates; branches with support < 50% were collapsed. For comparison, we performed an additional maximum parsimony analysis in TNT 1.5 [42], using 'New Technology' search options. Phylogenetic trees were visualized and edited in Fig-Tree 1.4.4 [43].

**Bayesian analysis.** For additional comparison, a Bayesian analysis in MrBayes 3.2.4 [44] under the MKv+G model [45] was conducted (see S1 and S2 Figs for details).

## Anatomical abbreviations

A, anal fin; aa, angulo-articular; abd vert, abdominal vertebrae; AP, anal fin pterygiophores inserting before haemal spine of first caudal vertebra; bh, basihyal; BH, maximal depth at onset of D1; BL, body length (horizontal line from first vertebral centrum to end of hypural plates); br, branchiostegal rays; C, principal caudal fin rays; caud vert, caudal vertebrae; ce, ceratohyal;

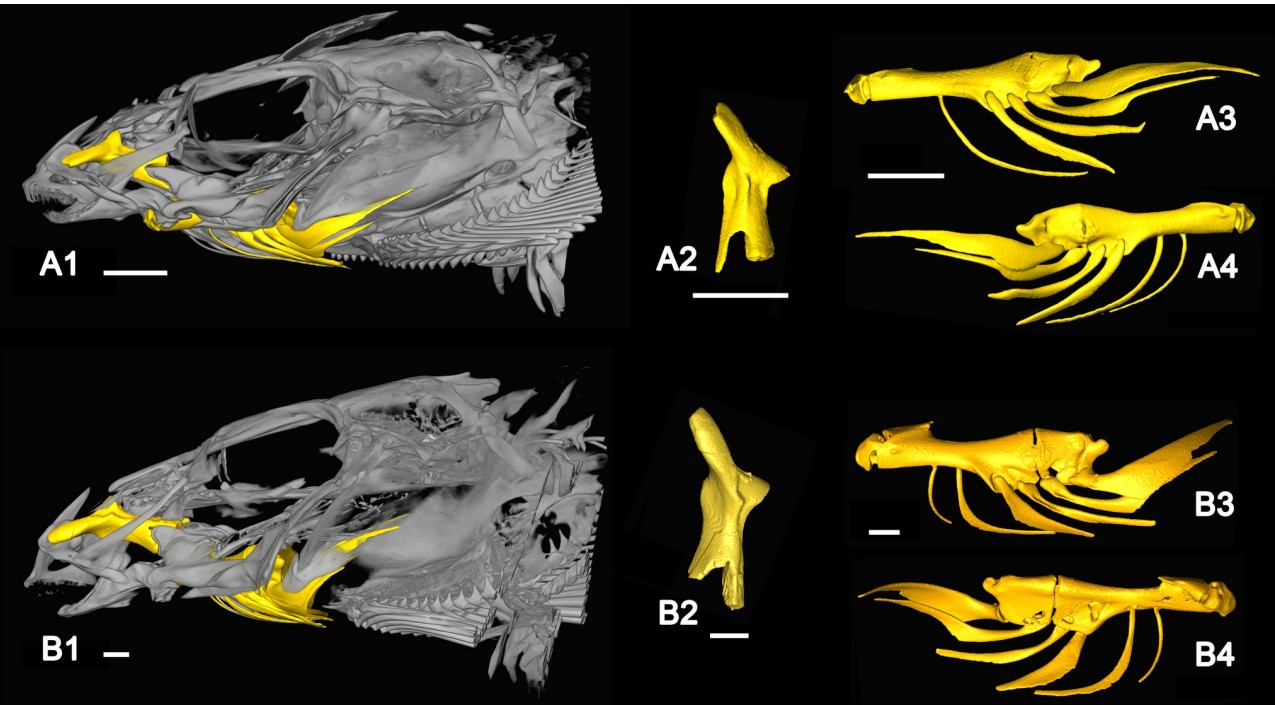

**Fig 4. Anatomical position and details of the palatine, the ceratohyal bone, and the branchiostegal rays of *Rhyacichthys guilberti*.** (A) MNHN 2019-0113-1, (A1) skull, (A2) left palatine, (A3) left ceratohyal with only five branchiostegal rays, (A4) right ceratohyal with six branchiostegal rays (all lateral view). (B) MNHN 2019-0113-2, (B1) skull, (B2) left palatine, (B3, B4) left ceratohyal in lateral (B3) and medial view (B4), each with six branchiostegal rays. All images based on μ-CT scanning. All scale bars 1 mm.

ce (a), anterior ceratohyal; ce (p), posterior ceratohyal; cl, cleithrum; CPH, caudal peduncle height (maximal depth of caudal peduncle); CPL, caudal peduncle length (horizontal line from end of anal fin to end of hypural plates); D1, first dorsal fin; D2, second dorsal fin; D2C, distance between end of D2 and first dorsal procurrent ray of caudal fin; den, dentary; ED, eye diameter (horizontally); ect, ectopterygoid; ent, entopterygoid; ethm, ethmoid; ethm pr pal, ethmoid process of palatine; fr, frontal; HL, head length; hm, hyomandibular; LPect, maximum length pectoral fin rays; LPelv, maximum length pelvic fin rays; mx, maxilla; mxl pr pal, maxillary process of palatine; orb, orbit; op, opercle; pal, palatine; Pect, pectoral fin; Pelv, pelvic fin; pmx, premaxilla; pop, preopercle; prC, procurrent caudal rays; psph, parasphenoid; ptt, posttemporal; r, radials; SL, standard length; SN/A, distance from tip of the snout to begin of A; SN/D1, distance from tip of the snout to begin of D1; SN/D2, distance from tip of the snout to begin of D2; sop, subopercle; sy, symplectic; vert, vertebrae; vo, vomer.

## Results

### The palatine bone of the extant species

The position of the palatine bone within the skull and its terminology are shown in Fig 3. In *Rhyacichthys guilberti*, the palatine head displays a very long maxillary process and a much smaller, angular ethmoid process (Fig 4A2 and 4B2). The palatine of *Protogobius attiti*, *Perccottus glenii* and *Eleotris pisonis* is more or less L-shaped (*sensu* Regan [10] and Hoese [26]), i.e. showing a large and relatively blunt maxillary process, but no clear ethmoid process (Fig 4A2, 4B2 and 4C2). In contrast, the palatine of *Gobius incognitus* is 'T-shaped' (*sensu* [10]), i.e. the

palatine head has two prominent processes, one for connection to the maxillary and the other one for articulation with the ethmoid (Figs 3B3 and 4D2).

## The branchiostegal rays of the extant species

One of the specimens of *Rhyacichthys guilberti* examined displays six branchiostegal rays at each hyoid bar. The second individual has six branchiostegal rays on the right (Fig 4A4), but only five on the left hyoid bar (Fig 4A3). The specimens of *Protogobius attiti*, *Perccottus glenii* and *Eleotris pisonis* each display six branchiostegal rays (Fig 5A3, 5A4, 5B3 and 5C3), while the specimen of *Gobius incognitus* reveals a number of five branchiostegal rays (Fig 5D3). In all studied specimens, the last branchiostegal ray is considerably expanded compared to the preceding ones and is articulated with the posterior ceratohyal bone (Fig 4). In *R. guilberti* the penultimate branchiostegal ray is also associated with the posterior ceratohyal (Fig 4A3, 4A4, 4B3 and 4B4), while in *P. attiti* the penultimate branchiostegal ray articulates at the anterior ceratohyal, but close to the gap that separates this part of the bone from the posterior part (Fig 5A3 and 5A4). In the examined specimen of *P. glenii* the position of the penultimate branchiostegal ray is not recognisable because the bone of the ceratohyal is partially dissolved (Fig 5B3). In *E. pisonis* (Fig 5C3) and *G. incognitus* (Fig 5D3) the penultimate branchiostegal ray is 'shifted' anteriorly (away from the gap) and articulates with the anterior ceratohyal.

## Systematic palaeontology

Teleostei Müller, 1845 *sensu* Arratia [46]
Percomorphaceae *sensu* Betancur-R. et al. [47]
Gobiiformes *sensu* Betancur-R. et al. [11]
Suborder Gobioidei Günther, 1880
Family †Pirskeniidae Obrhelová, 1961

### Genus †*Pirskenius* Obrhelová, 1961

*†Pirskenius diatomaceus Obrhelová, 1961*. Figs 6–9, Tables 3 and 4
1961 *Pirskenius diatomaceus* n. sp.–Obrhelová, p. 111, pl. I–XV, text-Figs 1–29.
2014 *Pirskenius diatomaceus* Obrhelová, 1961.–Přikryl: p. 188.
*Material*. New specimens NMP Pv 11669, 11671, 11672. Holotype NMP PC 2769 (no. 8 in [15]) and specimens NMP PC 2770–PC 2822.
*Provenance and age*. Knížecí (Czech Republic); early Oligocene.
*Preservation*. The holotype is a moderately well preserved imprint of a complete individual. Many of the further specimens are well preserved (some as part and counterpart), but most specimens are incomplete, lacking either the caudal or the anterior part of the body. The three new specimens (Pv 11669, 11671, 11672) are very well preserved and revealed several previously unknown characters of †*P. diatomaceus* (see below).
*General description*. The total length of †*P. diatomaceus* ranges between 13 and 60 mm. The head and the eyes are relatively large (HL 24% SL, EL 27–37% HL, Table 3; Figs 6A, 6C, 7, 9A and 9C1). The oral gape is terminal and opened (Fig 9A). The body is long and slender, tapering only slightly towards the caudal fin (BL 75.4–76.7% SL, BH 9–13% SL; Figs 6B–6D and 9C1). The paired fins are relatively short (13–14% SL), with the pectoral fin being slightly longer than the pelvic fin. The second dorsal and the anal fin are placed approximately midway along the body, with the anal fin inserting slightly behind the second dorsal fin (Figs 6B, 6C and 9A, 9C1). The bases of the second dorsal and the anal fin are relatively short (13–15% SL and 10–15% SL, respectively). The caudal peduncle is long and slender (CPL 24.6–33.7% SL,

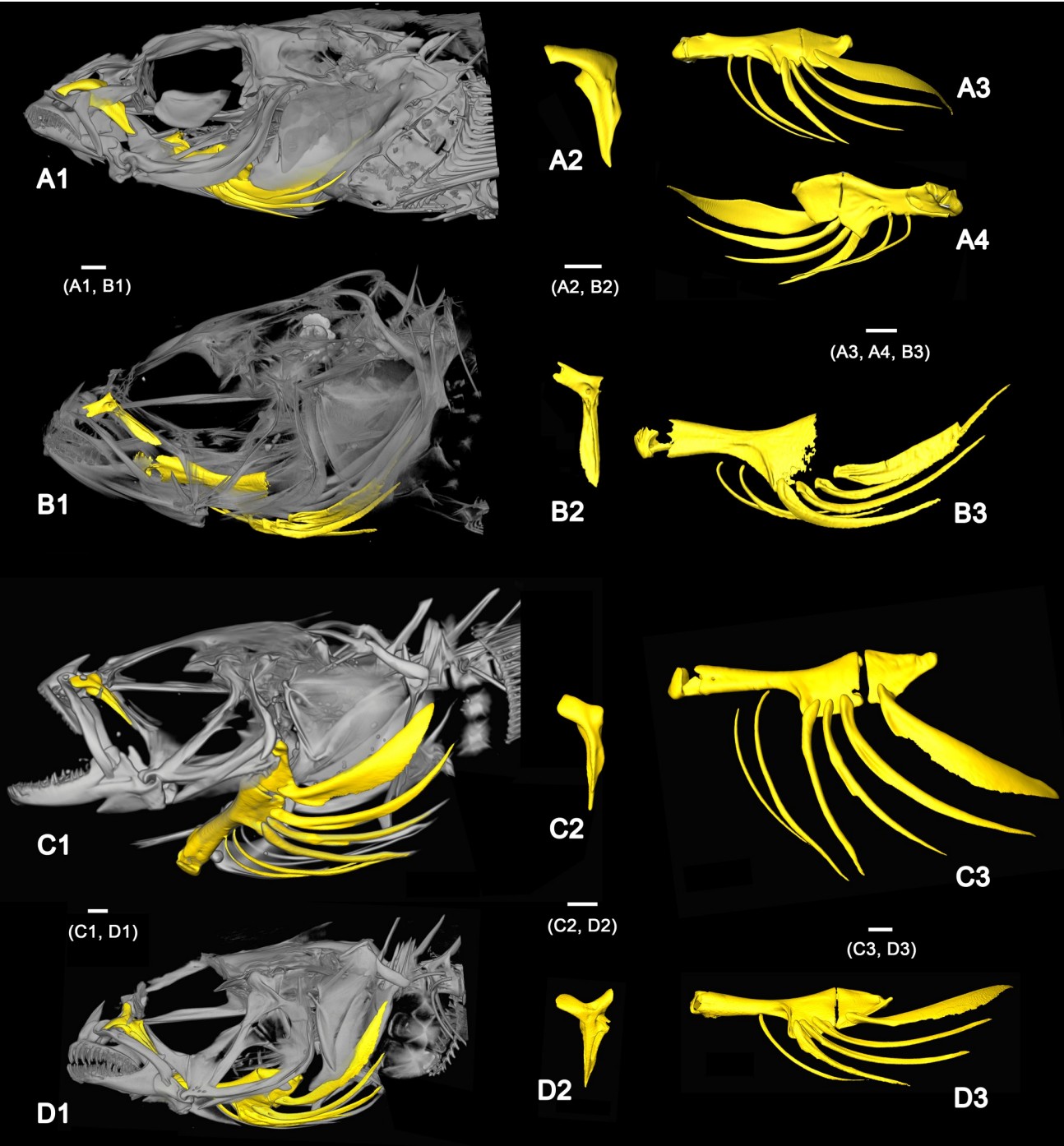

**Fig 5. Anatomical position and details of the palatine, the ceratohyal bone, and the branchiostegal rays of extant gobioid species.** (A) *Protogobius attiti* (MNHN 2019–0112), (A1) skull, (A2) left palatine, (A3, A4) left ceratohyal with six branchiostegal rays in lateral (A3) and medial (A4) view. (B) *Perccottus glenii* (UWFC 44788), (B1) skull, (B2) left palatine, (B3) left ceratohyal (bone only partially preserved) with six branchiostegal rays. (C) *Eleotris pisonis* (ZSM 9393), (C1) skull, (C2) left palatine, (C3) left ceratohyal with six branchiostegal rays. (D) *Gobius incognitus* (NMP6V 146150), (D1) skull, (D2) left palatine, (D3) left ceratohyal with five branchiostegal rays. All images based on µ-CT scanning, all images show lateral view except A4 (medial view). All scale bars 1 mm.

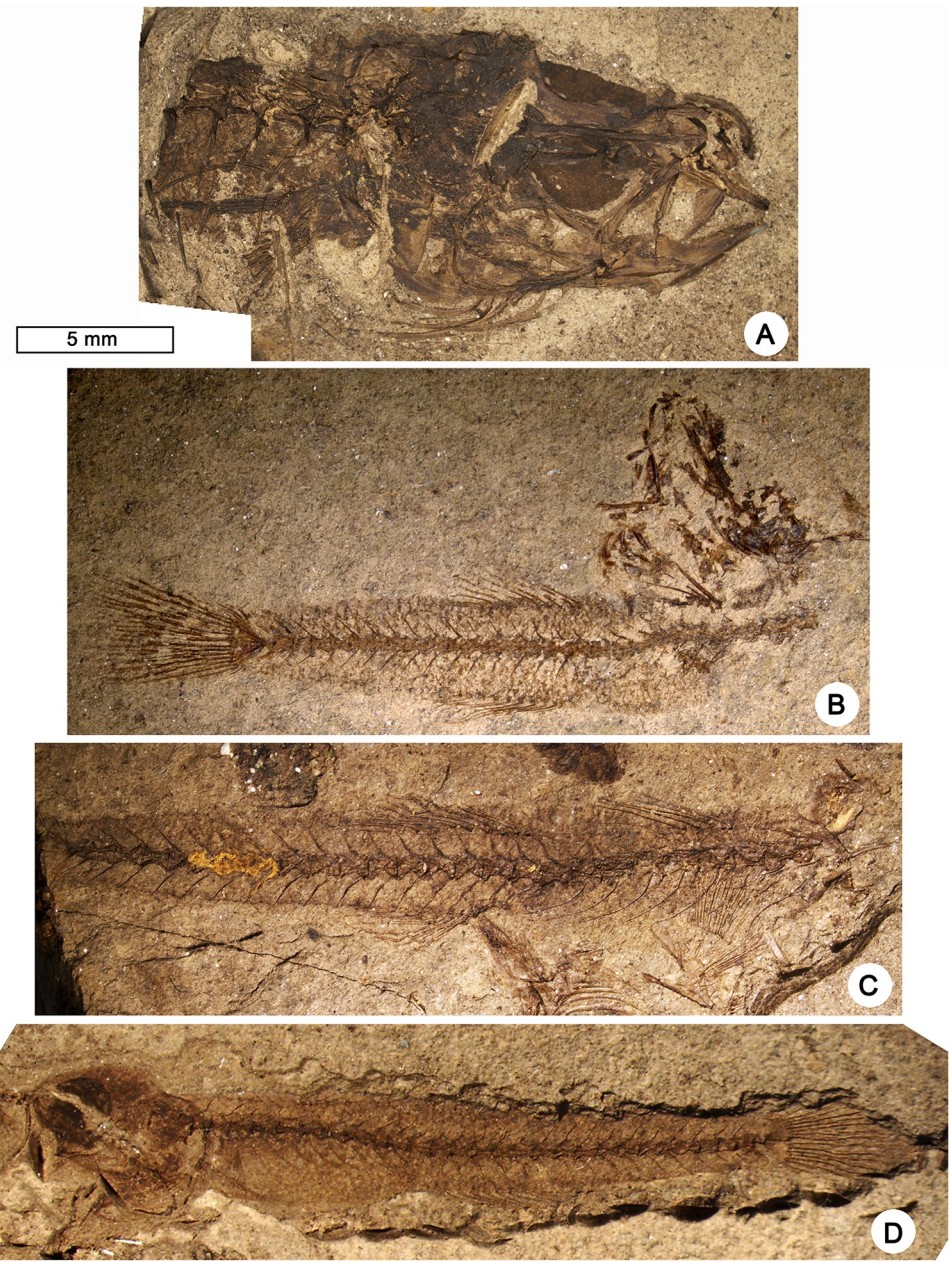

**Fig 6. Fossil skeletons of †*Pirskenius diatomaceus* Obrhelová, 1961.** (A) Specimen Pv 11669, skull (ventrolateral view) and anteriormost portion of the body (lateral view) exhibiting seven branchiostegal rays. (B) Specimen Pv 11672, complete skeleton with slightly disarticulated skull. (C) Specimen Pv 11671, skeleton with well-preserved dorsal fins, anal fin and caudal peduncle. (D) Specimen PC 2799, almost complete skeleton.

CPH 7.7–11.6% SL), and appears to be relatively longer in the largest specimens (Table 3). The caudal fin is fan-shaped to slightly rounded in form (Fig 6B).

*Head* (Figs 7 and 8). The frontal bones show the typical gobioid condition, with a trapezoid shape, a broad postorbital section and a slender interorbital segment (Fig 7). The width of the interorbital segment is about 21–25% of the width of the postorbital sector (PC 2786: 21.3%, PC 2787: 22.9%, PC 2791: 24.8%, Pv 11669: c. 25%); the value of 16% provided for specimen PC 2786 by Obrhelová ([15]: Table 14, as no. 22) could not be confirmed. The parasphenoid is

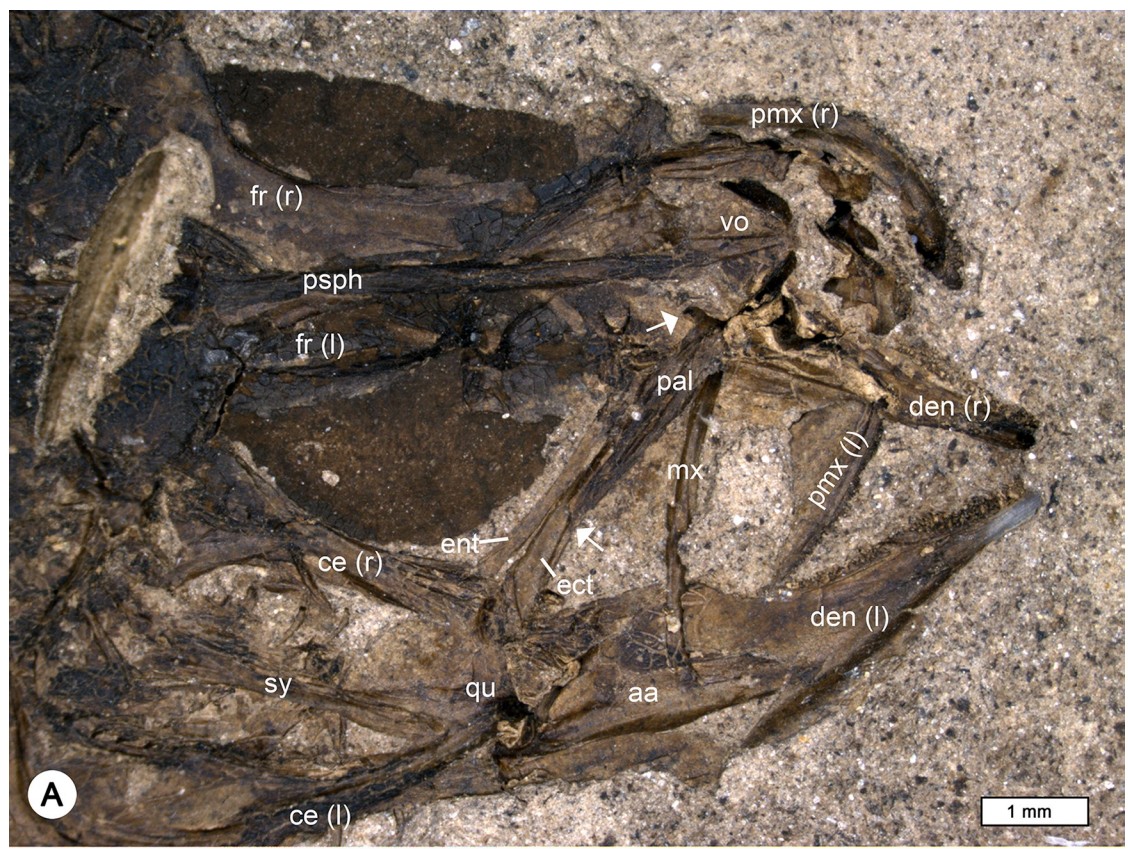

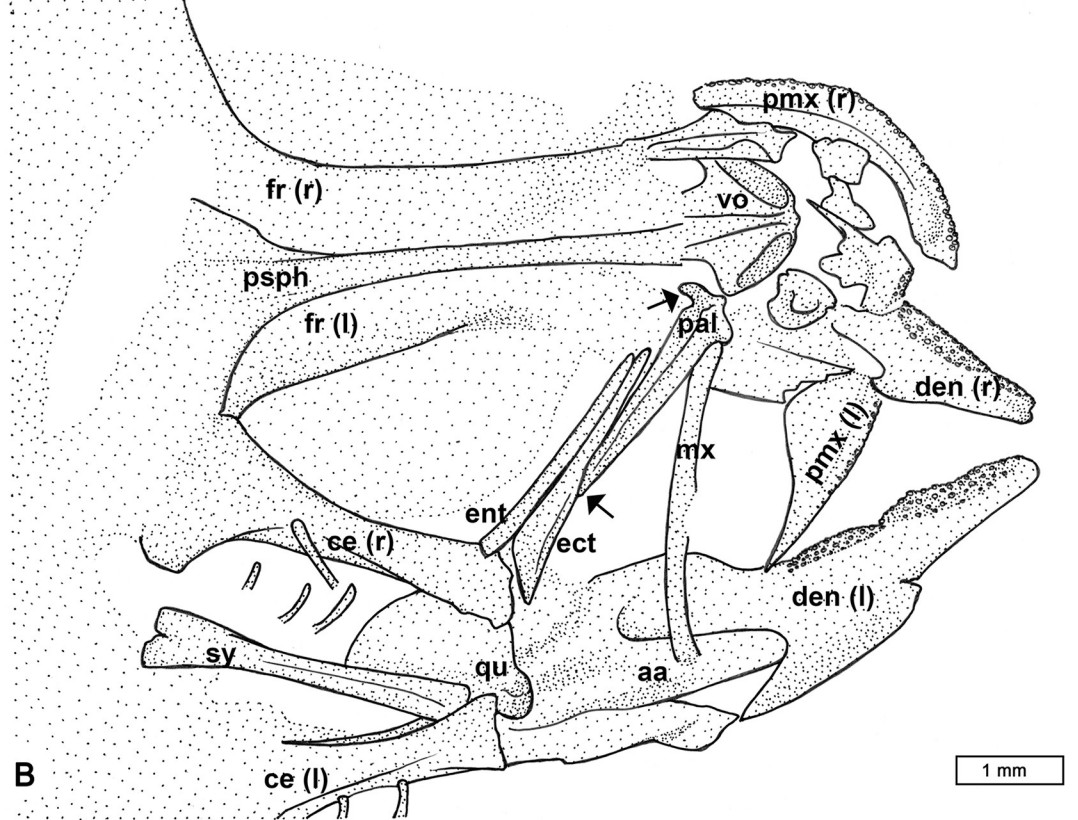

**Fig 7. Details of the head and jaw bones of †*Pirskenius diatomaceus* Obrhelová, 1961.** (A, B) Photo and reconstruction of specimen Pv 11669 (ventrolateral view) exposing bones of right (r) and left (l) side, with palatine (pal), ectopterygoid (ect) and entopterygoid (ent) preserved in anatomical connection. Upper arrow indicates ethmoid process of palatine, lower arrow points to ventral tip of palatine. Further abbreviations: aa, angulo-articular; ce, ceratohyal; den, dentary; fr, frontal; mx, maxilla; pmx, premaxilla; psph, parasphenoid; qu, quadrate; sy, symplectic; vo, vomer.

a long, slender bone, broadened only in the posterior region of the neurocranium; anteriorly it is associated with a broad, triangular vomer, which does not bear teeth (PC 2791, Pv 11669; Fig 7). The bones of the ethmoid and otic regions and of the occiput are not clearly discernible. Parietal bones and infraorbital bones are absent.

The jaw joint is located slightly in front of (PC 2787, PC 2804), or below the anteriormost portion of the orbit (PC 2707, PC 2772). The premaxilla has a relatively long ascending process and a somewhat shorter articular process. The horizontal ramus of the premaxilla bears a prominent, wedge-shaped postmaxillary process, which gradually tapers posteriorly, so that the posterior end of the premaxilla is slender (Fig 7, also visible in specimen PC 2786). The premaxilla bears small conical teeth that are arranged in several rows (well visible in specimen PC 2772). The maxilla is a slender, slightly curved bone (Fig 7), its articular process is usually obscured by other bones or not well preserved; according to Obrhelová [15], it is relatively long and forms a U-shaped articular facet for the articulation with the palatine. The dentary is relatively slender, and posteriorly bifurcated (Figs 7 and 9A). The relatively large head of Pv 11669 reveals details of the lower jaw dentition: its left dentary is preserved in labial view and comprises two different sectors (Fig 7), the anterior sector bears irregularly arranged clusters of teeth (sandpaper type), while the teeth in the posterior sector are organised in rows.

The palatine is a robust, straight, and relatively short bone, whose posteroventral tip ends approximately midway along the ectopterygoid (indicated by the lower arrow in Fig 7A and 7B). According to Obrhelová [15], the maxillary process of the palatine is straight and slender, continuing in the direction of the longitudinal axis of the palatine, but it is actually not visible in any of the specimens examined because other bones conceal it. Some specimens show the respective part of the palatine to be associated with the lateral side of the vomer, and it seems probable that Obrhelová [15] mistakenly identified the thickened anterolateral margin of the vomer as a long slender maxillary process of the palatine. Notably, the palatine head bears a short lateral ethmoid process (indicated by the upper arrow in Fig 7A and 7B). The ectopterygoid is elongate, approximately as long as the maxilla, and widens towards the quadrate (Fig 7). The entopterygoid appears to be as long as the ectopterygoid and is straight (Fig 7). The quadrate has a triangular body with a distinct articular head; anteriorly, it is firmly attached to the ventral margin of the ectopterygoid, posteriorly it has a long process that extends to the preopercle. The symplectic is a relatively robust rod that articulates with the posterior margin of the corpus quadrati, i.e. to the fossa quadrati (Fig 7). A broad gap between the symplectic and the preopercle (suspensorium fenestra or symplectical foramen) is clearly discernible (Fig 7). The slender metapterygoid is not connected to the quadrate. The hyomandibular is a relatively broad and somewhat rounded bone, with three distinguishable processes (Fig 7).

The hyoid bar consists of a relatively long anterior ceratohyal bone, which begins as a slender element and then broadens, and a relatively short, broad and triangular posterior ceratohyal bone (Fig 7); a small gap is visible between the two parts and could represent an originally cartilaginous region. The number of branchiostegal rays is always seven (Figs 6A and 8, clearly detectable also in PC 2774, PC 2775, PC 2791, PC 2798). The three anteriormost branchiostegal rays articulate with the slender portion of the anterior ceratohyal, and are thin and regularly spaced (Fig 8). After a gap follow the relatively thick and closely spaced branchiostegal rays 4–6, which are attached to the widened portion of the anterior ceratohyal. The

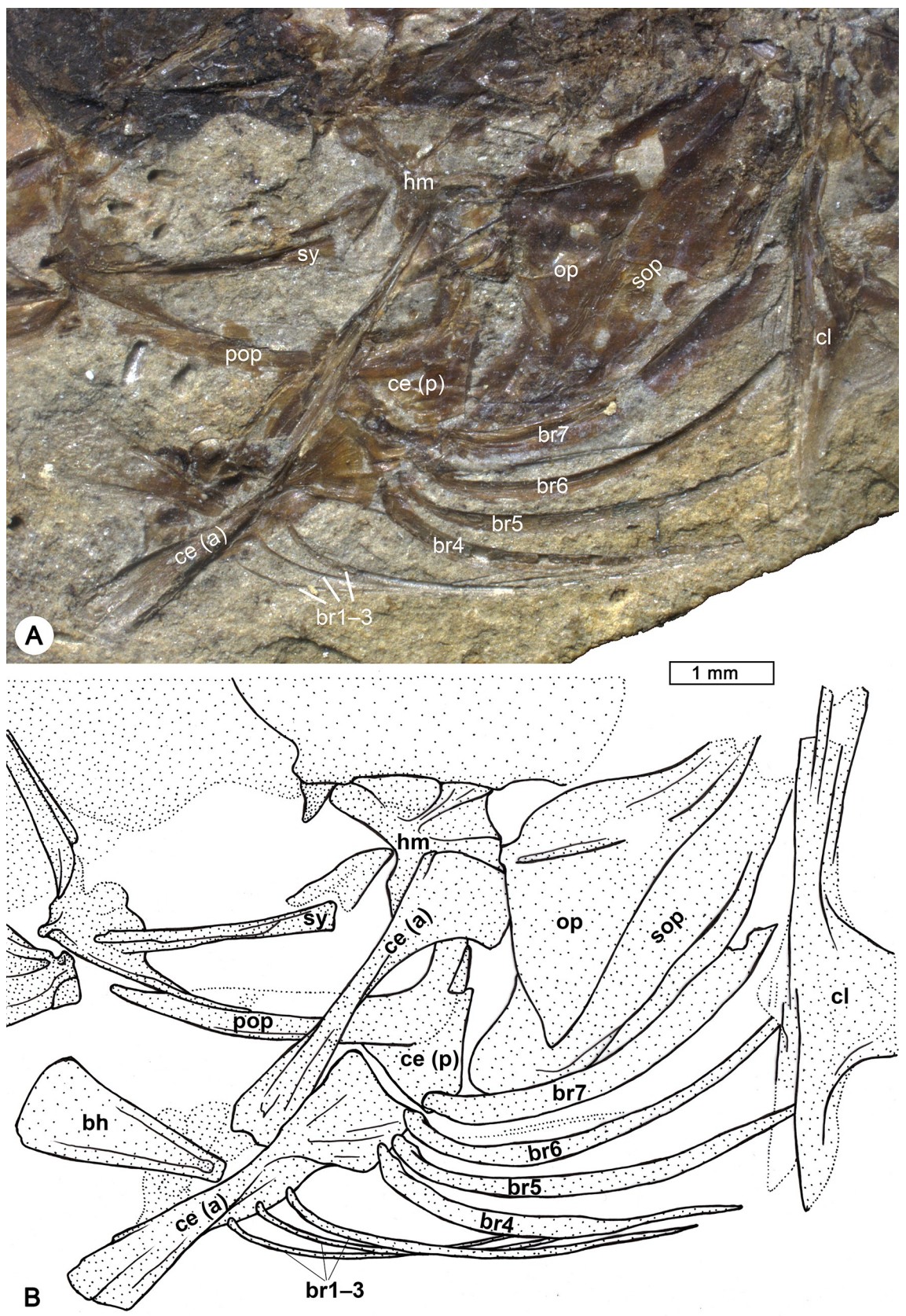

**Fig 8. Details of the branchiostegal rays and opercular bones of †*Pirskenius diatomaceus* Obrhelová, 1961.** (A, B) Photo and reconstruction of specimen PC 2786 showing branchiostegal rays (br) 1–6 articulating with anterior ceratohyal (ce (a)), and expanded branchiostegal ray 7 associated to posterior ceratohyal (ce (p)). Further abbreviations: bh, basihyal; cl, cleithrum; hm, hyomandibular; mx, maxilla; op, opercle; pop, preopercle; sop, subopercle; sy, symplectic.

sixth branchiostegal ray is located just before the gap between the anterior and posterior ceratohyal (Fig 8). The seventh branchiostegal ray is the broadest of the series and articulates with the posterior ceratohyal (Fig 8). The opercular bones are best preserved in specimens Pv 11669 (Fig 6A) and PC 2786 (Fig 8). The subopercle is elongate and crescent-shaped, the opercle is triangular, and the preopercle has a slender lower and an upper arm of almost equal length. Specimen Pv 11669 reveals a well preserved bony preopercular canal (*sensu* Hoese and Gill [4]) extending along the horizontal and vertical branch of the preopercle (Fig 6A).

*Vertebral column* (Fig 9B, Table 4). The number of abdominal vertebrae is 11, and there are 16–17 caudal vertebrae, with the exception of specimen PC 2791, which has 18. The neural spines are slender and of approximately equal length all along the vertebral column. Haemal spines are as long as neural spines, apart from that of the first caudal vertebra, which exhibits a somewhat shortened haemal spine, and preural vertebra 2, which has an enlarged haemal spine. The abdominal vertebrae have robust parapophyses, in some specimens their connection with the ribs is preserved. The total number of rib pairs is 8–9, beginning at vertebra 3 and extending to vertebra 10 or 11; the first seven rib pairs are robust and relatively long, the posteriormost pairs are thinner and shorter (Fig 9B, also well visible in PC 2799, Pv 11669, Pv 11671). Epipleurals are associated with the first four vertebrae (Pv 11669). There are no supraneurals.

*Caudal skeleton* (Figs 6B and 9C2, Table 4). The caudal skeleton is composed of two wide hypural plates (composed of hypurals 1+2 and 3+4), the fifth hypural plate is small and thin. A robust parhypural and a large, single epural are present. The caudal fin is fan-shaped. It usually comprises 13, rarely 14 principal rays, of which seven are placed in the dorsal section. The number of procurrent caudal rays is up to 14 dorsally, and up to 11 ventrally.

*Unpaired fins* (Figs 6B, 6C, 9A and 9B, Table 4). The first dorsal fin (D1) has six or seven spines, the last of which is set apart from the preceding spine by a small gap. The length of the spines decreases continuously from the first or second spine to the last. Each D1 spine is supported by a long, slender pterygiophore (indicated by the arrows in Fig 9B); only the first pterygiophore is broadened distally. The first pterygiophore inserts behind the neural spine of vertebra 3 and the last pterygiophore behind the neural spine of vertebra 6 or 7 (depending on whether six or seven D1-spines are developed). Between the last D1 pterygiophore and the first D2 pterygiophore is an interneural space without a pterygiophore. The D1 pterygiophore formula is recognisable in four specimens, it is 3(122110) in the case of seven D1 spines, and 3 (12210) in the case of six spines.

The second dorsal fin (D2) begins slightly in front of the insertion of the anal fin. The first element is a slender spine, which is slightly shorter (i.e. about four fifths) than the following ray. The total number of rays is nine to ten. All rays are segmented and branched; the anterior rays are long, the more posterior rays become increasingly shorter. The first D2 pterygiophore is a slender bone and probably bipartite, i.e. comprising a long proximal and a relatively shorter distal radial, but no middle radial (specimen PC 2772).

The anal fin inserts beneath the first or second ray of D2. It consists of a slender spine, which is somewhat shorter than the following nine to ten rays. The number of anal fin pterygiophores that insert anterior to the haemal spine of the first caudal vertebra is two (specimens PC 2773, Pv 11672) or three (PC 2791).

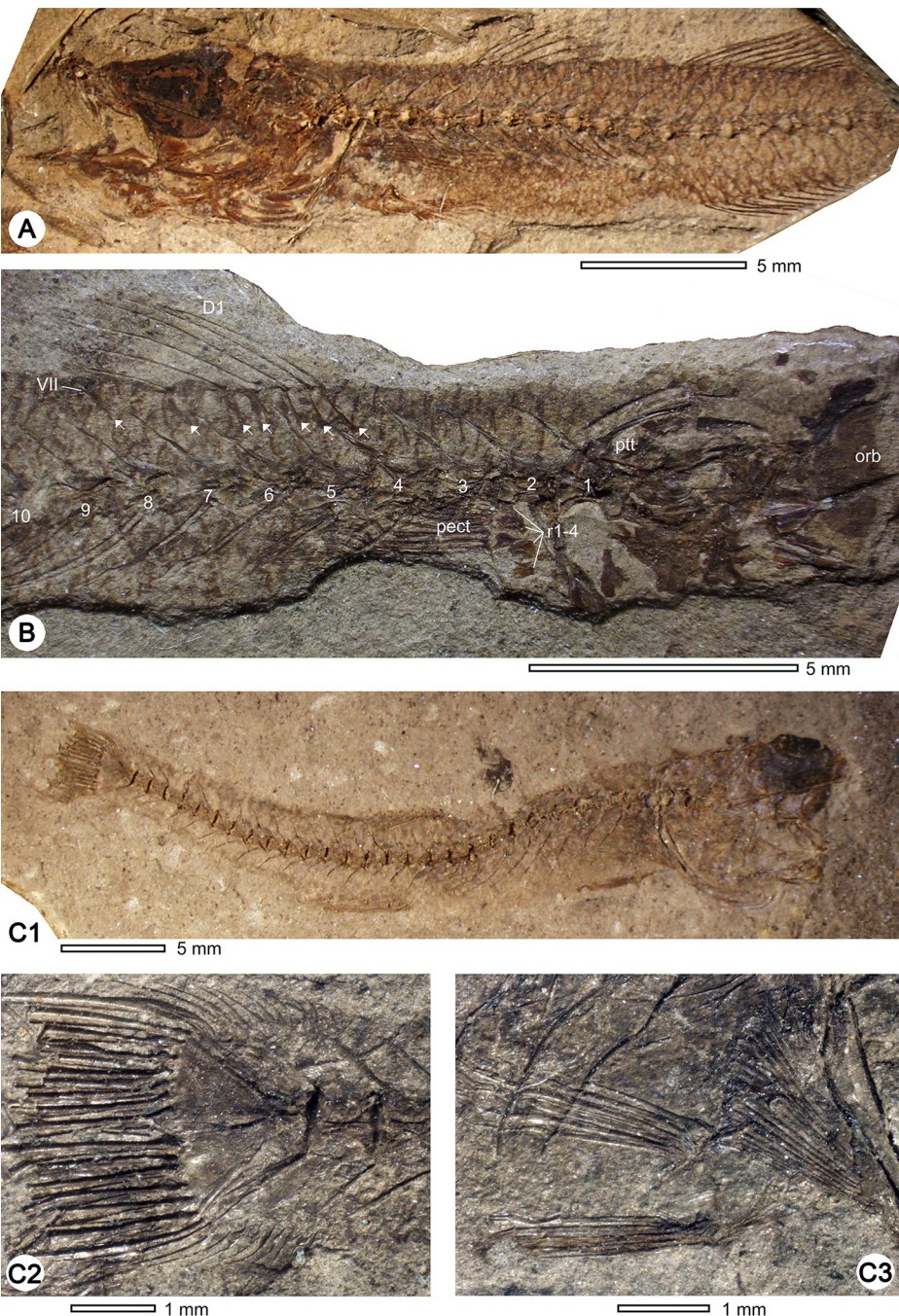

**Fig 9. Fossil skeletons of †*Pirskenius diatomaceus* Obrhelová, 1961 showing details of fins.** (A) Specimen PC 2772 exhibiting dorsal and anal fins (posterior part not preserved). (B) Counterpart of (A) (PC 2773) displaying radials (r1–r4) and remains of the pectoral fin (pect) and first dorsal fin with D1 pterygiophore formula 3(122110) (pterygiophores indicated by arrows). (C1–3) Specimen PC 2791, overview (C1), close-up of caudal fin with well-preserved procurrent rays (C2), close-up of separated pelvic fins (C3). Further abbreviation: ptt, posttemporal.

*Paired fins and their skeleton* (Fig 9B and 9C3, Tables 2 and 3). The cleithrum is long, with a slender dorsal part and a widened ventral portion; the dorsal margin of the cleithrum has a 'cleithral notch' sensu Winterbottom [29] (Fig 8B). The supracleithrum is robust and associated with a bifurcated posttemporal, the dorsal process of which is longer than the ventral (Fig

**Table 3. Morphometric data of †*Pirskenius diatomaceus*.**

| | PC 2777 | PC 2813 | Pv 11672 | PC 2787 | PC 2794/2795 | PC 2799 | Pv 11671 | PC 2779 | PC 2772 | PC 2791 | Pv 11669 |
|---|---|---|---|---|---|---|---|---|---|---|---|
| **SL** | 23.2 | 23.4* | 24.9 | 26.2* | 30.5 | 30.5 | 34.1* | 35.3* | 35.8* | 36.6* | 55.4* |
| **HL** | 5.4 (23.3) | | 6.0 (24.0) | 6.3 | 7.2 (23.6) | 7.5 (24.6) | | | 8.6 | (8.8) | 13.3 |
| **ED** | 2.0 (37.0) | | | 1.9 (30.1) | 2.7 (37.5) | 2.4 (32.0) | | | 2.9 (33.7) | 2.4 (27.3) | 3.6 (27.0) |
| **BL** | 17.8 (76.7) | | 18.9 (75.9) | | 23.3 (76.4) | 23 (75.4) | 25.9 | 26.8 | | 27.8 | |
| **BH** | 2.2 (9.4) | | c. 3.2 (12.8) | | 3.5 (11.5) | 3.9 (12.8) | | 3.5 (9.9) | 4.7 (13.1) | c. 4.5 (12.3) | |
| **SN/D1** | - | | 9.1 (36.5) | | | 11.6 (38.0) | | | 13.2 (36.9) | | |
| **SN/D2** | 12.4 (53.4) | | 13.7 (55.0) | | c. 16 (c. 52.4) | 16.8 (55.1) | | | 19.8 (55.3) | | |
| **SN/A** | 13.6 (58.6) | | 14.2 (57.0) | | c. 17.1 (56.1) | 17.4 (57.0) | | | 20.8 (58.1) | | |
| **LPect** | | | | | | | | | c. 5.0 (14.0) | | |
| **LPelv** | | | | | | | | | 4.5 (12.6) | | |
| **D2 base** | | 3.6 (15.4) | 3.5 (14.0) | | | | 4.6 (13.5) | 4.6 (13.0) | 4.5 (12.6) | c. 4.9 (13.4) | |
| **A base** | | 3.5 (14.9) | 3.3 (13.2) | | 3.1 (10.2) | 3.3 (10.8) | 4.0 (11.7) | 4.2 (11.9) | 4.2 (11.7) | 3.7 (10.1) | |
| **CPL** | 5.7? (24.6) | 7.5 | 7.2 (28.9) | | c. 9.8 (32.1) | 9.9 (32.4) | 11.5 (33.7) | 11.8 (33.4) | | 11.8 (32.2) | |
| **CPH** | 1.8 (7.7) | 2.7 (11.5) | 2.9 (11.6) | | 3.1 (10.2) | 2.9 (9.5) | c. 3.2 (9.4) | 3.8 (10.8) | | 3.3 (9.0) | |
| **CPL/ CPH** | 3.1 | 2.75 | 2.5 | | 3.2 | 3.4 | 3.6 | 3.1 | | 3.6 | |
| **D2C** | | 7.3 (31.2) | 6.6 (26.5) | | 9.7 (31.8) | | 10.0 (29.3) | 10.2 (28.9) | | 9.0 (24.5) | |

Specimens are arranged according to their standard length (SL). Values depict measurements in mm, values in brackets refer to % SL, only in case of eye diameter (ED) values refer to % of head length (HL).

\* indicates that SL was calculated based on measurement of body length (BL) or HL or caudal peduncle length (CPL) by assuming a proportion of BL = 76% of SL, HL = 24% of SL, and CPL = 32% SL, respectively, based on measurements of complete specimens. For abbreviations see Material and methods.

9B). There is no postcleithrum and no scapula. The pectoral fins insert on the lower one-third of the flank and are relatively short (c. 14% SL). Each fin is supported by four elongate radials with ovate-shaped gaps in between (also visible in PC 2785/2786). According to Obrhelová [15], the number of pectoral fin rays is 15–16, rarely 12, 14, or 17 rays. However, a number of 12–14 pectoral fin rays is clearly displayed by specimens PC 2804, PC 2791 and Pv11671 and thus appears to represent the correct condition. The pelvic girdle is a robust and medially fused bone; its anterior part is slightly rounded to heart-shaped, the posterior portion pointed and triangular. The pelvic fins are inserted just under or slightly behind the pectoral fins and are clearly separated. Each pelvic fin includes one spine and five rays. The rays are slightly shorter (12.6% SL) than those of the pectoral fin, and clearly terminate anterior to the anal fin.

*Scales*. Ctenoid scales cover the flanks (Fig 9A and 9B). Because they overlap, only their somewhat thickened posterior margins are visible; radii are not preserved. A single row of

**Table 4.  Meristic data of †*Pirskenius diatomaceus*.**

| | PC 2771 | PC 2772/2773 | PC 2774 | PC 2779 | PC 2781 | PC 2788 | PC 2791 | PC 2794/2795 | PC 2796 | PC 2799 | PC 2804 | PC 2809 | Pv11671 | Pv11672 |
|---|---|---|---|---|---|---|---|---|---|---|---|---|---|---|
| **Abd vert** | | 11 | | 11 | 11 | 11 | 11 | 11 | 11 | | | | 11 | 11 |
| **Caud vert** | | | | 17 | 17 | 18 | 17 | 17 | 17 | | | | 16 | 16 |
| **D1** | VI | VII | VII | VII | VI | | | VII | | | VI | | VII | VII |
| **D2** | | I9 | | | I10 | | | I10 | | | I10 | | I9 | I9 |
| **D1 pt-formula 3 (122110)** | | + | + | + | | | | | | | | | | |
| **D1 pt-formula 3(12210)** | | | | | | + | | | | | | | | |
| **Anal fin** | | I9 | | | I10 | | | | | I9 | | | I9 | I10 |
| **AP** | | 2 | | | | | 3 | | | | | | | 2 |
| **Pectoral fin** | | | | | | | | | | | 12–13 | | 13–14 | |
| **Pelvic fin** | | | I5 | | | | | | | | I5 | | I5 | |
| **Caudal fin (dorsal/ventral)** | | | | | 7/6 | 7/6 | 7/6 | 7/6 | 7/7 | | 7/6 | | | 7/6 |
| **Dorsal prC** | | | | | >7 | 11 | 12 | >7 | >7 | | 14 | | 11 | 11 |
| **Ventral prC** | | | | | 9 | 11 | 8 | 11 | c. 8 | | 10 | | | 9 |

Specimens are arranged according to their collection numbers. For abbreviations see Material and methods.

relatively long and slender ctenii is present at the posterior scale margin. Scale width can be estimated to be 1.5–1.7% SL (measured below the second dorsal fin and close to the vertebral column). Transverse rows include 5–6 scales, longitudinal rows about 60. The hypural plates are covered with up to eight relatively rounded ctenoid scales with well preserved circuli. The head is scale-less. The lateral line is absent.

## †*Pirskenius radoni* Přikryl, 2014

2014 *Pirskenius radoni* n. sp.; Přikryl: 189, Figs 1–4.

*General description.* †*Pirskenius radoni* has similar body proportions as †*P. diatomaceus*, except that its head is slightly larger and the body is less elongate (Table 5). The scales of this species are not preserved. A re-examination of the holotype (MT PA1480) and paratypes (MT PA1482, 1483) revealed some additional details, which are reported in the following. For a complete and detailed description see Přikryl [17].

*Head.* The preserved remains of the premaxilla suggest the presence of a relatively high postmaxillary process, as in *P. diatomaceus*. The shape of the ectopterygoid resembles a strongly elongated triangle (rather than the 'L' shape depicted in Přikryl [17]). The endopterygoid is stick-like and located dorsally to the ectopterygoid. The palatine extends along half the length of the ectopterygoid, as in †*P. diatomaceus*. Přikryl [17] noted a clear maxillary process of the palatine and concluded that the palatine is L-shaped. However, the anterior part of the palatine is not clearly discernible. Preserved fragments of skeletal tissue suggest the presence of the maxillary process, but whether the ethmoid process was developed is not clear, because the vomer covers the area in which it would be expected. Since a short ethmoid process is present in †*P. diatomaceus*, and the palatine configuration is unlikely to vary between congeneric species, we assume that a short ethmoid process was also present on the palatine of †*P. radoni*; thus the palatine of †*P. radoni* was not exactly L-shaped.

The anterior ceratohyal bone shows a slightly concave dorsal margin and a strongly concave ventral margin; its anterior part is relatively narrow and it becomes about twice as deep in the

**Table 5. Comparison of morphometric data (in % of SL) and meristic counts between the two species of †*Pirskenius*.**

| | †*P. diatomaceus* | †*P. radoni* |
|---|---|---|
| Standard length | 23.2–55.4 | 17.8 \| 34.0 |
| Head length | 23.3–24.6 | 28.7 \| 32.4 |
| Body height | 9.4–13.1 | - \| 19.4 |
| SN/D1 | 36.5–38.0 | 45.5 \| 39.4 |
| SN/D2 | 52.4–55.3 | - \| 59.4 |
| SN/A | 57.0–58.1 | 62.4 \| 59.7 |
| Caudal peduncle length | 28.9–33.7 | - \| 25.3 |
| Caudal peduncle height | 7.7–11.6 | 7.9 \| 9.1 |
| Branchiostegal rays | 7 | 7 |
| Abdominal vertebrae | 11 | 12 |
| Caudal vertebrae | 16–17(18) | 16 |
| First dorsal fin | VI–VII | VII |
| Second dorsal fin | I9–10 | I8 |
| D1 pterygiophore-formula 3(122110) | + | |
| D1 pterygiophore-formula 3(12210) | + | |
| D1 pterygiophore-formula 4(32110) | | + |
| Anal fin | I9–10 | I9 |
| AP | 2–3 | 4 |
| Pectoral fin | 12–14 | >12 |
| Pelvic fin | I5 | I5 |
| Procurrent caudal rays (dorsal/ventral) | 7/6 (7/7) | 8/6 |

Ranges and counts for †*P. diatomaceus* refer to the values provided in Tables 3 and 4, data for †*P. radoni* originates from Přikryl [17] and this study. For abbreviations see Material and methods.

posterior section. Seven branchiostegal rays are present. In size, shape, and configuration, they resemble those of †*P. diatomaceus* (see above).

*Caudal skeleton.* The caudal skeleton of †*P. radoni* is composed of the same elements as in †*P. diatomaceus* (see Přikryl [17]). A single epural with a clearly discernible suture is present, suggesting that two epural bones were fused. Compared to †*P. diatomaceus*, the caudal fin of †*P. radoni* is more rounded (see Přikryl [17]). It is composed of 14 principal rays (eight dorsally and six ventrally, not 7 + 7 as reported by Přikryl [17]). Several dislocated procurrent rays are recognisable dorsally and ventrally, probably more than five in each case.

*Unpaired fins.* The first dorsal fin is composed of seven spines. The most probable D1 pterygiophore formula is 4(32110), as the first pterygiophore seems to be located posterior to the neural spine of the fourth vertebra (rather than after the third, as noted in the original description). The first D1 spine is shorter than the succeeding ones, apart from the seventh spine, which is the shortest. As in †*P. diatomaceus*, the seventh spine is separated from the preceding ones by a small gap. The second dorsal fin is composed of one spine and eight rays. The spine has more or less the same length as the adjacent rays, which become shorter posteriorly. The anal fin is composed of a single spine and nine rays. The spine is approximately as long as the anteriormost ray, the subsequent rays become gradually smaller posteriorly. The number of anal fin pterygiophores is four.

*Paired fins and their skeleton.* The pectoral girdle, which is only partially preserved, includes an arch-shaped cleithrum (that is somewhat straighter in its ventral portion), which is connected to the posterior part of the skull via the supracleithrum and a V-shaped posttemporal (note that the interpretative drawing in Přikryl [17]: fig 4 incorrectly depicts the latter elements as fused). The ventral limb of the posttemporal is significantly shorter than the dorsal. The ventral part of the coracoid is more or less discernable.

## Discussion

### †*Pirskenius* in light of the new data

Obrhelová [15] introduced the family †Pirskeniidae with the single genus †*Pirskenius*. She provided a detailed diagnosis of both family and genus, together with a very comprehensive description of †*P. diatomaceus*, the sole species of †*Pirskenius* known at that time. Her work is excellent in many respects, especially when one considers that very little was known about the osteology of gobioids when she began her study. Nevertheless, there are some inconsistencies and obscurities in her work, some of which were previously noted by Springer [48], Nelson [49], Gaudant [19] and Přikryl [17]. These concern mainly the (1) number of branchiostegal rays, (2) presence or absence of entopterygoid, (3) presence of ctenoid or cycloid scales, (4) number of abdominal vertebrae, and (5) terminology of the caudal skeleton. In the present study, these issues could be resolved.

1. In the original diagnosis of †Pirskeniidae and †*Pirskenius*, a number of six to seven branchiostegal rays is mentioned, but the species description refers exclusively to seven rays. Here, we confirm the presence of seven branchiostegal rays in all specimens of †*P. diatomaceus* and †*P. radoni* examined.

2. The entopterygoid is reported as 'present or absent' in the original family diagnosis and species description, but a 'very narrow' entopterygoid is mentioned in the original genus diagnosis. According to the results of our study and Přikryl [17], an entopterygoid is present in both †*P. diatomaceus* and †*P. radoni*.

3. The scales are reported as 'ctenoid or cycloid' in the family diagnosis, but as 'ctenoid' in the genus diagnosis. The re-investigation of †*P. diatomaceus* revealed the presence of exclusively ctenoid scales.

4. The number of abdominal vertebrae was reported as '11 (12)' in the genus diagnosis of †*Pirskenius*. In contrast, a number of exclusively 11 was mentioned in the species description, which is also consistent with the results of this study. Twelve abdominal vertebrae are found only in †*P. radoni* ([17] this study).

5. The terminology in use for the caudal skeleton at the time Obrhelová [15] conducted her study differs from that employed today [48, 50]. Judging from her drawings and photos of the caudal skeleton of †*P. diatomaceus* in Obrhelová [15] and the results of this study, it is clear that her 'two upper Epiuralia' correspond to hypural plate 5 and the epural, and that her 'lower Epiuralia' represents the parhypural. This is the common condition of the caudal skeleton in gobioids [51, 52].

Moreover, some previously unknown characters of †*Pirskenius* could be discerned in the present study, mostly based on the new specimens of that genus. These relate to the postmaxillary process of the premaxilla, the teeth on the dentary, the shape of the palatine head, the bony preopercle canal support, the articulation of the sixth branchiostegal ray, and the presence of an interneural gap between the first and second dorsal fin (see above). In addition, the

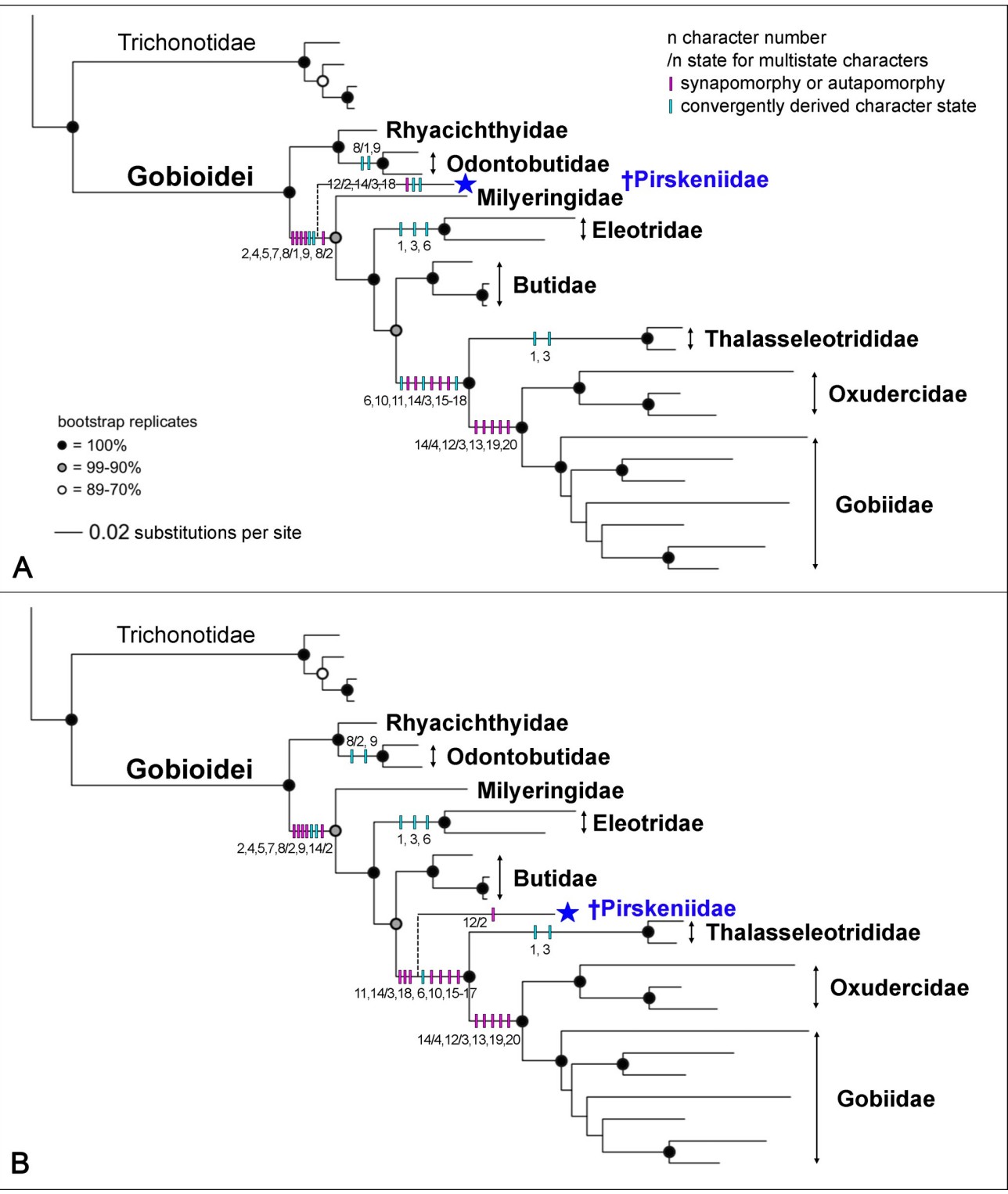

**Fig 10. Two possible phylogenetic positions of the †Pirskeniidae based on mapping of morphological characters on a recently published molecular tree of extant Gobioidei.** (A) †Pirskeniidae is sister to all extant gobioid families except Rhyacichthyidae and Odontobutidae. (B) †Pirskeniidae is sister to Thalasseleotrididae + Gobiidae + Oxudercidae. Synapomorphies or autapomorphies are indicated with pink bars, convergently derived character states are shown in light blue, multistate characters are followed by a slash (/) after which the character state is indicated. For character numbers see Table 2. Tree adapted from Thacker et al. [7] 'based on DNA sequences of ten nuclear protein coding genes with a taxon sampling expanded from Near et al. [62]'.

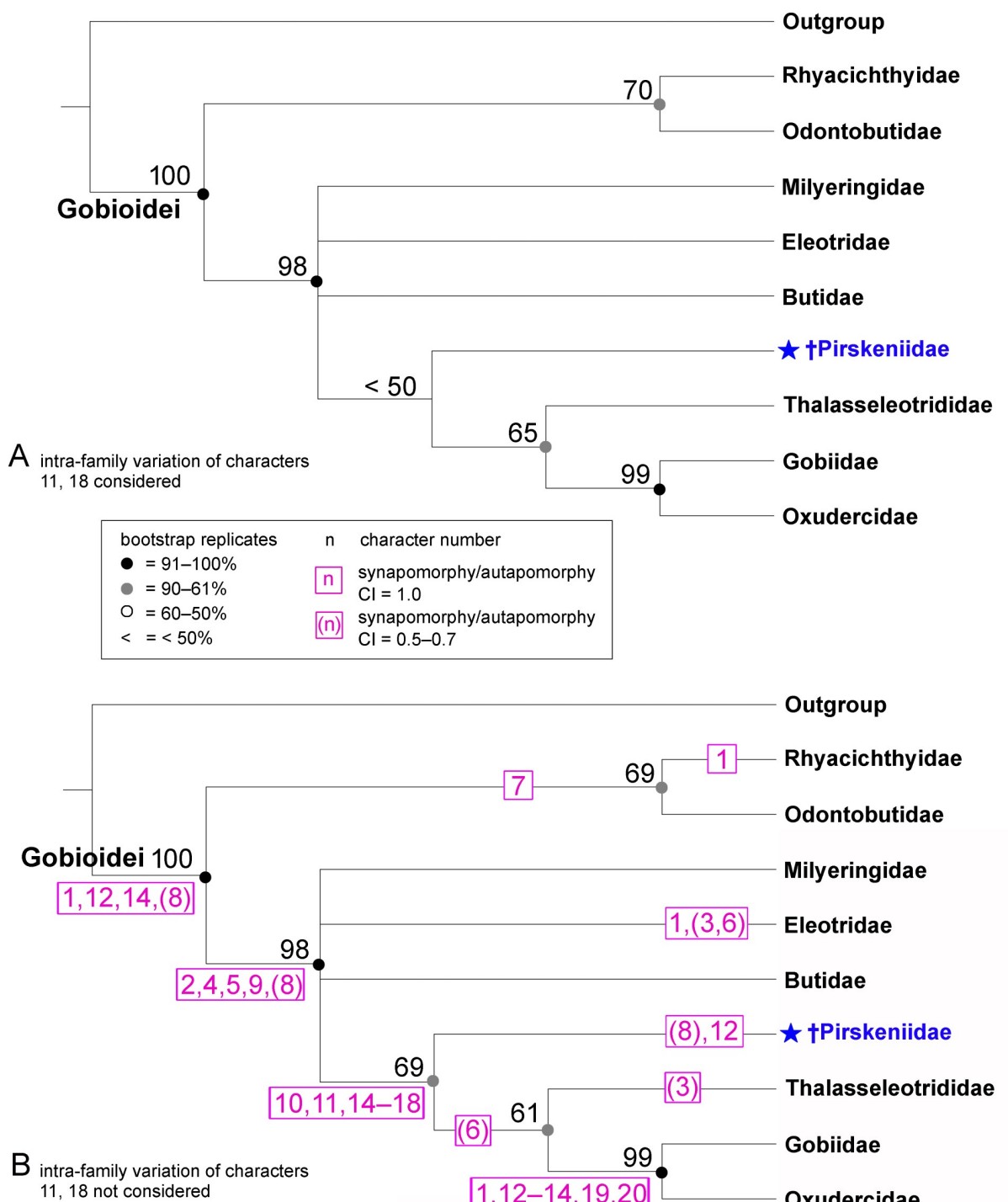

**Fig 11. Phylogenetic position of the †Pirskeniidae based on maximum-parsimony analysis of 20 phylogenetically informative morphological characters.** (A) 50% majority rule consensus tree based on 20 most parsimonious trees inferred with PAUP*; matrix includes intra-family variation of characters 11 and 18 (see Table 2 for details). (B) 50% majority rule consensus tree based on 14 most parsimonious trees inferred with PAUP*; matrix excludes intra-family variation of characters 11 and 18. For both trees, tree length = 32 steps, CI = 0.906, RI = 0.914. Numbers in boxes are synapomorphies (respectively autapomorphies) as indicated by PAUP*; synapomorphies are the same for both trees. Numbers at nodes are bootstrap percentages from 1000 pseudoreplicates.

D1 pterygiophore formula, which was first introduced by Birdsong [53], could be established. Unfortunately, it sheds no light on the taxonomic position of †*Pirskenius*. Of the two formulas detected for †*P. diatomaceus*, the formula 3(12210) commonly occurs in the Oxudercidae, occasionally in some eleotrids, and (as an exception) in the butid *Kribia* (see [21]), whereas the formula 3(122110) is only known from some eleotrids (*Philypnodon*, *Dormitator*, *Gobiomorphus*), the odontobutid *Perccottus* and (as an exception), from the gobiid *Psilotris* (see [21]). The formula 4(32110) (†*P. radoni*) is unknown among extant gobioids ([21] and unpublished data of BR).

Hereafter, a revised diagnosis of the genus †*Pirskenius* is presented. It excludes several of the characters provided in the original genus diagnosis because, according to the current knowledge of the osteology of gobioids, they appear in many groups. These characters relate to the anteriorly elongate and posteriorly broadened parasphenoid; the toothed premaxillary and dentary; the presence of small, curved, conical oral teeth; the symplectic foramen; the opercle without spines; the bifurcated posttemporal; the presence of four radials; the presence of vertebral centra that are longer than high; the caudal skeleton with two large hypural plates (Hy), comprising Hy1+2 and Hy3+4, and a small hypural plate 5; and the presence of branched fin rays (see [4–6, 20, 29, 48, 51, 52, 54–56]). The shape of the frontal bones is also excluded, as it has been shown to differ between congeneric species [12]. A non-toothed vomer, although occurring in many groups of gobioids, is not excluded because one species of the Eleotridae, *Eleotris vomerodentata* Maugé, 1984, endemic to Madagascar, was described as new because of plenty teeth on the vomer [57].

**Revised diagnosis of †*Pirskenius* Obrhelová, 1961.** Medium-sized gobioid fish, with the following unique combination of characters: up to 6 cm total length, with typical gobioid placement of the median and paired fins and long caudal peduncle (25–34% of SL); head relatively large (HL 23–32% SL); non-toothed vomer; premaxilla with distinct postmaxillary process; dentary relatively slender; entopterygoid present; palatine with short, slender ethmoid process; seven branchiostegal rays; postcleithrum absent, scapula absent; separated pelvic fins; 11–12 abdominal vertebrae; total number of vertebrae 27–28, rarely 29; well-developed parapophyses; pectoral fin rays 12–14; D1 VI–VII; D2 I8–10; A I9–10; two to four anal fin pterygiophores inserting anterior to haemal spine of first caudal vertebra; 13–14 principal caudal fin rays; single epural.

## The family †Pirskeniidae–Valid or not?

Gaudant [19] and Přikryl [17] had suggested that the combination of characters presented as diagnostic for the †Pirskeniidae is not unique among the Gobioidei. Both authors considered the palatine of †*Pirskenius* to have an eleotrid shape (*sensu* Regan [10]) and synonymised †Pirskeniidae with the Eleotridae. However, a number of seven branchiostegal rays, as in †*Pirskenius*, is unknown among extant Gobioidei [20]. This number occurs in their next close relatives, the Trichonotidae, Apogonidae and Kurtidae [58–60], but the members of these families lack the expansion of the last branchiostegal ray as is characteristic for the Gobioidei [38]. Přikryl [17] regarded the presence of seven branchiostegal rays in †*Pirskenius* as an ancestral state of the Eleotridae, because Akihito [8] had reported one specimen of *Odontobutis obscurus* (Temminck and Schlegel, 1845) as showing the same condition. However, this observation refers to an apparently anomalous character state, as the other 11 specimens examined by Akihito had six branchiostegals. The comparative material used here also includes one instance of a similar irregularity: in one of the specimens of *Rhyacichthys guilberti* the left hyoid bar has five branchiostegal rays (Fig 4A3), instead of the normal condition of six rays. As *Rhyacichthys* represents an early-branching gobioid [4, 5], and as five branchiostegal rays is the condition of

the modern Gobiidae and Oxudercidae [9], this finding also can be treated as an exception. Occasional deviations in the number of branchiostegal rays can thus occur as random cases, without phylogenetic significance. As a consequence, the presence of seven branchiostegal rays sets †Pirskeniidae apart from all extant gobioid families and represents an autapomorphy for this family. However, as an autapomorphy, it is of no help in establishing the phylogenetic relationships of the taxon.

The second crucial character of †*Pirskenius* is the occurrence of a palatine head with a short and slender ethmoid process (Fig 7) and it is probable that †*Pirskenius* did not have a truly T-shaped palatine (*sensu* Regan [10]) as is characteristic for the Gobiidae and Oxudercidae [26, 61]. Nonetheless, the condition of a palatine with a slender ethmoid process is derived as it occurs in the Thalasseleotrididae [23], but is not present in the Odontobutidae, Eleotridae and Butidae [8, 26, 28]. †Pirskeniidae exhibits further derived characters, namely: uppermost radial of pectoral fin adjoins cleithrum, absence of ossified scapula, absence of dorsal postcleithrum, no lateral line, presence of interneural gap between last pterygiophore of first dorsal fin and first pterygiophore of second dorsal fin, and presence of a single epural (see above and Table 2). It is thus justified to resurrect the family.

**Revised diagnosis of †Pirskeniidae Obrhelová, 1961.** Medium-sized gobioid fish, up to 6 cm total length, with seven branchiostegal rays and the typical gobioid placement of the median and paired fins. Entopterygoid present; palatine head with short slender ethmoid process; uppermost radial of pectoral fin adjoins cleithrum; scapula absent; postcleithrum absent; separated pelvic fins; ctenoid scales without transforming ctenii.

## Phylogenetic relationships of †Pirskeniidae

To analyse the possible phylogenetic position of the †Pirskeniidae we used two approaches: (i) derived states of morphological characters (see Table 2) were plotted on the molecular tree of the Gobioidei of Thacker et al. [7] and the †Pirskeniidae were added to this tree manually as suggested by the character state distributions; (ii) a phylogenetic analysis was performed under maximum parsimony in PAUP* (as well as TNT) using the character matrix described in the Methods section (Table 2, characters 1–20); for comparison also a Bayesian analysis in MrBayes was conducted (see Methods for details).

**Character mapping on a molecular phylogeny.** Character mapping suggests two options to place †Pirskeniidae within the molecular tree of the Gobioidei provided by Thacker et al. [7]: †Pirskeniidae is sister to all Gobioidei, except Rhyacichthyidae and Odontobutidae (Fig 10A), or †Pirskeniidae is sister to Thalasseleotrididae + Gobiidae + Oxudercidae (Fig 10B). In the first option, particular weight is laid on the penultimate branchiostegal ray position (see states of character 8 in Table 2). The plesiomorphic condition is exhibited by *Rhyacichthys*, in which the penultimate branchiostegal ray is placed at the posterior ceratohyal (= state 0; see Fig 4A3, 4A4, 4B3 and 4B4). In †Pirskeniidae the penultimate branchiostegal ray articulates at the anterior ceratohyal, but near to the gap that separates this part of the bone from the posterior ceratohyal (= state 1, see Fig 8). Such a condition is also known for the Odontobutidae [4], and was here also found in *Protogobius attiti* (member of the Rhyacichthyidae) (Fig 5A3 and 5A4). This could indicate a placement of †Pirskeniidae close to the Rhyacichthyidae and Odontobutidae. All other families are more specialised insofar as the penultimate ray articulates with the anterior ceratohyal, clearly anterior to the gap to the posterior ceratohyal (= state 2; see Fig 5C3 and 5D3). However, †Pirskeniidae reveals three further characters that preclude a placement near the Odontobutidae, namely an uppermost radial that adjoins the cleithrum, while the scapula is absent (see states of character 4 in Table 2), and presence of ctenoid scales that lack transforming ctenii (see states of character 7 in Table 2). These three characters

support monophyly of all gobioids (including †Pirskeniidae) except Rhyacichthyidae and Odontobutidae (see [4] and Fig 10). Only with respect to the exact position of the penultimate branchiostegal, †Pirskeniidae is set apart from the remaining families, as indicated in Fig 10A.

The second option of the character mapping approach is not using the exact position of the penultimate branchiostegal ray as described above, but follows Hoese and Gill [4]. Accordingly, the derived condition is that the penultimate branchiostegal ray articulates with the anterior ceratohyal, irrespective of its exact position. In this case, †Pirskeniidae displays the derived condition and does not split off early as shown in Fig 10A. Option 2 is further supported because †Pirskeniidae displays a palatine that has a short slender ethmoid process and thus is not exactly L-shaped, and it lacks the dorsal postcleithrum (see states of characters 14 and 18 in Table 2). The palatine shape would be unique for †Pirskeniidae and the clade Thalasseleotrididae + Oxudercidae + Gobiidae, and the absent dorsal postcleithrum would remain a synapomorphy in the precise definition of Gill and Mooi [9] (Fig 8B) (although exceptions are known, see Hoese [26]). In the alternative scenario of option 1 (Fig 10A) characters 14 and 18 would both require convergent evolution in †Pirskeniidae and Thalasseleotrididae + Oxudercidae + Gobiidae. Thus option 2, with Thalasseleotrididae + Oxudercidae + Gobiidae being sister to †Pirskeniidae, is the most parsimonious interpretation of the phylogenetic relationships of the family.

**Maximum parsimony analysis.** Phylogenetic analysis of the extant gobioid families and the †Pirskeniidae was conducted based on 20 parsimony-informative morphological characters (Table 2). In a first step, we considered the intra-family variation of all characters, as presented in Table 2 (see also Supporting Information S1 File). The corresponding PAUP* analysis inferred 20 most parsimonious trees, each with a length of 31 steps and a relatively low degree of homoplasy (CI = 0.903, RI = 0.914); the 50% majority rule consensus tree is shown in Fig 11A. In a second step, we removed from our character matrix the occasional presence of an interneural gap (character 11) in the Eleotridae and Butidae [21], and the occasional absence of the dorsal postcleithrum (character 18) in the Eleotridae [26] (indicated in squared brackets in Table 2, see also Supporting Information S2 File). Now the PAUP* analysis inferred 14 most parsimonious trees, tree length, CI and RI were the same as in the previous analysis; the 50% majority rule consensus tree is shown in Fig 11B. The topology of both trees (Fig 11A and 11B) is the same; †Pirskeniidae is recovered as sister to the Thalasseleotrididae + Gobiidae + Oxudercidae clade, consistent with option 2 above (Fig 10B). Notably, although the two PAUP* analyses are based exclusively on morphological characters, the results are relatively congruent with molecular phylogenies (e.g. [3, 7], see Fig 10); but the exact positions of Milyeringidae, Eleotridae and Butidae are not resolved. The tree that excluded the intra-family variation of characters 11 and 18 (Fig 11B) revealed moderate bootstrap support (69%) for the node leading to the †Pirskeniidae (vs. bootstrap support was < 50% in the other tree).

For comparison, a corresponding phylogenetic analyses, i.e. considering all intra-family variation in the matrix vs. excluding intra-family variation of characters 11 and 18, was conducted using TNT and MrBayes, the results are shown in the S1 and S2 Figs. †Pirskeniidae also forms a clade with the Thalasseleotrididae + Gobiidae + Oxudercidae clade, with low bootstrap supports of <50 and 56 (S1A and S1B Fig) but notably high posterior probability of 0.97 and 0.98 (S2A and S2B Fig), but its exact position is not resolved. This was also the case in the strict consensus tree of the two PAUP* analyses (not shown), pointing to the possibility that †Pirskeniidae might be more closely related either to Thalasseleotrididae or to Gobiidae + Oxudercidae. However, more data need to be collected to discriminate between these hypotheses. Taking together, all phylogenetic analysis, i.e. character mapping on the molecular tree of the Gobioidei, maximum parsimony analysis as well as Bayesian analysis support the

position of the †Pirskeniidae as sister to the Thalasseleotrididae + Gobiidae + Oxudercidae clade.

## Conclusion

Here, we have established that †*Pirskenius diatomaceus* and †*P. radoni* represent species of the extinct family †Pirskeniidae, and are not members of the extant family Eleotridae. Previously reported fossil remains of putative eleotrids from the Oligocene and lower Miocene of Europe all refer to isolated bones [14, 19] or otoliths [63–65]. As isolated bones do not allow one to differentiate between Eleotridae and Butidae (see Fig 10), and as the 'eleotrid' otoliths actually represent otoliths of the Butidae [18], there is currently no evidence that eleotrids were present in the area at that time. Furthermore, our phylogenetic analyses suggest that †Pirskeniidae, which was restricted to the early Oligocene (ca. 29–30 Ma) of Central Europe, is phylogenetically close to the extant clade Thalasseleotrididae + Gobiidae + Oxudercidae.

## Supporting information

**S1 Fig. Phylogenetic position of the †Pirskeniidae based on maximum-parsimony analysis of 20 phylogenetically informative morphological characters using TNT.** (A) 50% majority rule consensus tree based on 5 most parsimonious trees; matrix includes intra-family variation of characters 11 and 18 (see Table 2 for details). (B) 50% majority rule consensus tree based on 4 most parsimonious trees; matrix excludes intra-family variation of characters 11 and 18. For both trees, tree length = 33 steps, CI = 0.879, RI = 0.886. Tree search was conducted using a combination of 'New Technology' search options (parsimony ratchet, tree-drifting, tree-fusing) under equal weighting of characters. In the strict consensus tree of (A), the node leading to the Thalasseleotrididae and †Pirskeniidae had collapsed, and the two families were placed in a polytomy with Milyeringidae, Eleotridae and Butidae; the strict consensus tree of (B) had the same topology as the 50% majority rule consensus tree (trees not shown). Numbers in boxes are synapomorphies (respectively autapomorphies) as indicated by TNT; see Table 2 for character descriptions. Numbers at nodes are bootstrap percentages from 1000 pseudoreplicates.
(TIF)

**S2 Fig. Phylogenetic position of the †Pirskeniidae based on Bayesian analysis of 20 phylogenetically informative morphological characters using MrBayes.** (A) 50% majority rule consensus tree based on 180,000 sampled trees from 2 independent runs; matrix includes intra-family variation of characters 11 and 18 (see Table 2 for details). (B) 50% majority rule consensus tree based on 180,000 sampled trees from 2 independent runs; matrix excludes intra-family variation of characters 11 and 18. For both trees the first 10% of trees were discarded as burn-in, chain length $10^6$ generations, 1 cold and 3 heated chains per run, sampling every 10th generation. Numbers at nodes are posterior probabilities; scale bar indicates number of expected character changes per character.
(TIF)

**S1 File. This is the Nexus file of the matrix used for the phylogenetic analyses shown in Fig 11A, S1A and S2A Figs.**
(NEX)

**S2 File. This is the Nexus file of the matrix used for the phylogenetic analyses shown in Fig 11B, S1B and S2B Figs.**
(NEX)

## Acknowledgments

We are grateful to Zlatko Kvaček (Charles University, Prague, Czech Republic) for providing the new specimens of †*Pirskenius*. We thank B. Ekrt (National Museum, Prague, Czech Republic) and M. Radoň (Museum of Teplice, Czech Republic) for access to the specimens in their care and for the opportunity to use the Keyence microscope at the Palaeontological Department of the National Museum in Prague. We also thank U. Schliewen and D. Neumann (both SNSB-ZSM, Munich, Germany) for providing access to specimens kept in the SNSB-ZSM collection, and B. Ruthensteiner (SNSB-ZSM) for technical support concerning visualization of micro-CT scans. We thank J. Vukić (Charles University, Prague, Czech Republic) and R. Šanda (National Museum, Prague, Czech Republic) for personally delivering the specimen of *Gobius ignotus*. We are indebted to H. Larson (Museum and Art Gallery of the Northern Territory, Darwin, Australia) and P. Chakrabarty (LSU Museum of Natural Science, Baton Rouge, Louisiana, USA) for providing information on the number of branchiostegal rays in *Milyeringa*. We thank the Vanuatu Environment Unit for granting access to *Rhyacichthys* (Permit Numbers ENV326/001/1/07/DK and ENV326/001/1/08/D). The Willi Hennig Society is acknowledged for making TNT available free of charge. We are grateful to D. Hoese (Australian Museum, Sydney, Australia) and an anonymous reviewer for their constructive reviews and valuable comments that greatly helped to improve our manuscript.

## Author Contributions

**Conceptualization:** Bettina Reichenbacher, Tomáš Přikryl.

**Data curation:** Bettina Reichenbacher, Tomáš Přikryl, Alexander F. Cerwenka.

**Formal analysis:** Bettina Reichenbacher, Tomáš Přikryl, Christoph Gierl, Martin Dohrmann.

**Funding acquisition:** Bettina Reichenbacher, Tomáš Přikryl.

**Investigation:** Bettina Reichenbacher, Tomáš Přikryl.

**Methodology:** Bettina Reichenbacher, Tomáš Přikryl, Alexander F. Cerwenka, Christoph Gierl, Martin Dohrmann.

**Project administration:** Bettina Reichenbacher, Tomáš Přikryl.

**Resources:** Bettina Reichenbacher, Tomáš Přikryl, Alexander F. Cerwenka, Philippe Keith.

**Software:** Bettina Reichenbacher, Tomáš Přikryl.

**Supervision:** Bettina Reichenbacher, Tomáš Přikryl.

**Validation:** Bettina Reichenbacher, Tomáš Přikryl.

**Visualization:** Bettina Reichenbacher, Tomáš Přikryl, Alexander F. Cerwenka.

**Writing – original draft:** Bettina Reichenbacher, Tomáš Přikryl.

**Writing – review & editing:** Bettina Reichenbacher, Tomáš Přikryl, Alexander F. Cerwenka, Philippe Keith, Christoph Gierl, Martin Dohrmann.

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
