## [Decision Letter · Decision Letter 0]

8 Jun 2020

PONE-D-20-14246

Freshwater gobies 30 million years ago: new insights into character evolution and phylogenetic relationships of †Pirskeniidae (Gobioidei, Teleostei)

PLOS ONE

Dear Dr. Reichenbacher,

Thank you for submitting your manuscript to PLOS ONE. After careful consideration, we feel that it has merit but does not fully meet PLOS ONE’s publication criteria as it currently stands. Therefore, we invite you to submit a revised version of the manuscript that addresses the points raised during the review process.

We look forward to receiving your revised manuscript.

Kind regards,

Giorgio Carnevale, Ph.D

Academic Editor

PLOS ONE

Additional Editor Comments:

Both the reviewers considered the manuscript an interesting contribution but also evidenced a number of issues that must be necessarily addressed.

2. Please include your tables as part of your main manuscript and remove the individual files. Please note that supplementary tables should be uploaded as separate "supporting information" files.

'We acknowledge funding for this project from the Deutsche Forschungsgemeinschaft to B.R. (RE-1113/20). T.P.’s research was institutionally supported by the Czech Academy of the Sciences, Institute of Geology (RVO67985831).'

'he funders had no role in study design, data collection and analysis, decision to publish, or preparation of the manuscript.'

4. We note that Figure 2 in your submission contains map images which may be copyrighted.

We require you to either (a) present written permission from the copyright holder to publish this figure specifically under the CC BY 4.0 license, or (b) remove the figure from your submission:

b. If you are unable to obtain permission from the original copyright holder to publish this figure under the CC BY 4.0 license or if the copyright holder’s requirements are incompatible with the CC BY 4.0 license, please either i) remove the figure or ii) supply a replacement figure that complies with the CC BY 4.0 license. Please check copyright information on all replacement figures and update the figure caption with source information. If applicable, please specify in the figure caption text when a figure is similar but not identical to the original image and is therefore for illustrative purposes only.

5. Your ethics statement must appear in the Methods section of your manuscript. If your ethics statement is written in any section besides the Methods, please move it to the Methods section and delete it from any other section. Please also ensure that your ethics statement is included in your manuscript, as the ethics section of your online submission will not be published alongside your manuscript.

Reviewers' comments:

Reviewer's Responses to Questions

**Comments to the Author**

1. Is the manuscript technically sound, and do the data support the conclusions?

Reviewer #1: Partly

Reviewer #2: Partly

2. Has the statistical analysis been performed appropriately and rigorously? 

Reviewer #1: No

Reviewer #2: N/A

3. Have the authors made all data underlying the findings in their manuscript fully available?

Reviewer #1: No

Reviewer #2: No

4. Is the manuscript presented in an intelligible fashion and written in standard English?

Reviewer #1: Yes

Reviewer #2: Yes

5. Review Comments to the Author

Reviewer #1: Authors have done a comprehensive job redescribing the species of Pirskenius and it is good to see an attempt to relate the fossils to existing taxa.

It is not clear why authors used Rhyacichthys. Protogobius or an Odontobutid head scan might have been more informative of primitive conditions. Rhyacichthys is a highly specialised gobioid. Protogobius has different palatine form Rhyacichthys. Some data for Protogobius is in Shibukawa et al.

With such a large group, authors have selected on 3 taxa to compare the fossil genus, which effectively rules out other groups and makes the character states, which are complex, look overly simple.

Figure legends are scattered throughout the text, which is confusing.

Figures 5 & 6 are hard to interpret. A drawing showing main points in text would be helpful.

Table 2. Pterygiophore formula broken into several characters. Primitive appear to be variable and specialisation fixed “mostly” to one state. Essentially a multistate character is being broken into several characters. Also, pterygiophore for Oxudercidae appears incorrect and should be 3-12210 – “mostly”, but see below. Character state for Eleotridae not given, variable but often 3-1221, but in some cases variable within a genus (Allomogurnda and Mogurnda). Appears selective for data analysis. One might argue that the 3(1221) of eleotrids gave rise to oxudercines 3(12210) and that butid 3(2211) gave rise to gobiids 3(22110). In other words different interpretations or codings of character states could give very different results.

Also, it is not clear what usually means in terms of number of species and genera. Also some odontobutids have the same pattern as Pirskenius with lots of variation.

Character 1. Adductor mandibular tendon –Hoese & Larson reported the condition in Thlasseleotris as the same as butines, but Hoese & Gill reported the condition as like eleotrids because of the lack of the process on the premaxilla; they did note the tendon in front of the ligament as in butines, although to a lesser degree. Also unfortunately figure 1a & b are reversed in their paper. Reexamination of Thalsseleotris and Grahamichthys indicates a condition more like butines or intermediate, although a subjective calling. In effect it probably should be a character state between 2 and 3, with no process, but tendon above middle of premaxilla.

Character 3 – no data presented. Hoese & Gill mentioned 3 states – not extending over epural, extending over anterior part of epural (in Thalasseleotris and Tateurndina), covering full epural (this state in eleotrids and some gobiids)

Character 6 – butid Kribia lacks canals and supports

Character 9 – Lateral line present in Terateleotris, considered to be an odontobutid.

Character 11. Interneural gap is present in Rhyacichthyids, some odontobutids, some butids (some Bostrychus) and some eleotrids, although unclear if homologous with gobiid, oxudercid, thalasseleotridid condition.

Character 14. Highly subjective and over simplified. Considerable variation is known to exist in the palatine shape (See Hoese 1984, Harrison, 1989). Gosline also 1958 found variation. Akihito 1969 also showed butine with T-shaped palatine. Palatine is L-shaped in Protogobius, but differently-shaped in Rhyacichthys. Not clear what is meant by inconstant in outgroup.

Character 18 Dorsal postcleithrum absent in some eleotrids.

Character 21 - Closure of first gill slit. – Not unique to Thlasseleotrididae as discussed by Gill & Mooi

Characters 22-25 are really not autapomorphies as they occur in various taxa in other groups. Autapomorphies are defined as unique. Defining a character state as usually is not unique.

Character 22. Blind eleotrids, gobiids and oxudercids are known. Lack of eyes – not particularly good character. Lack of eye relates to living in caves. Probably not phylogenetically important. Most caves are far too recent to affect long term evolution.

Character 23. Not really polarized. Authors are in effect saying every character state that doesn’t agree with the presumed specialisation is the primitive condition. Also based on assumption that everything has 6 dorsal spines, which doesn’t work for temperate gobioids.

Character 24 over 100 new world species have 3(221110). Also as for 23.

About half of eleotrine genera have 3(1221), but Tateurndina3(121111) or 3(122101), Allomogurnda, Gobiomorphus and Mogurnda highly variable within a species and other genera with 7 spines.

Character 25 – “amblyopine” often have 3(1221), Rhinogobius, Tridentiger and some European genera 3(22110), supposedly an apomorphy for gobiids. Many north Pacific species have various patterns such as 122010, 12111010, etc. Character in table is missing something – should be 3(12210). Also as for 23.

Hoese and Gill did not fully resolve the phylogeny and placement of dwarf gobioids, such as Kribia, Thalasseleotris, Grahamichthys, Tateurndina and dward “Oxyeleotris”, which did not have clear association with character states in other genera.

Authors should point out that gobioids are highly variable in most characters and discuss implication of that variation and how it could affect their analyses. Much of the variation is discussed in papers cited by the authors.

Authors also might want to comment of Carlomonnius, a supposedly gobioid from Eocene of Italy.

Reviewer #2: Thank you for the opportunity to review, please can you look at the attached PDF document with all of my detailed comments for this review added, this was a complex paper to review.

Many Thanks

6. PLOS authors have the option to publish the peer review history of their article (what does this mean?). If published, this will include your full peer review and any attached files.

Reviewer #1: Yes: Douglass Hoese

Reviewer #2: No

---

## [Author Response · Author response to Decision Letter 0]

21 Jul 2020

Please see Response to Reviewer comments for answers to all reviewer and editor comments.

---

## [Editor Report · Decision Letter 1]

24 Jul 2020

Freshwater gobies 30 million years ago: new insights into character evolution and phylogenetic relationships of †Pirskeniidae (Gobioidei, Teleostei)

PONE-D-20-14246R1

Dear Dr. Reichenbacher,

We’re pleased to inform you that your manuscript has been judged scientifically suitable for publication and will be formally accepted for publication once it meets all outstanding technical requirements.

Kind regards,

Giorgio Carnevale, Ph.D

Academic Editor

PLOS ONE

Additional Editor Comments (optional):

The authors did a considable job in improving the manuscript and iconography. The main issues raised by the reviewers have been addressed. It is my opinion that the manuscript is now suitable for publication in PLOS ONE.

---

## [Editor Report · Acceptance letter]

12 Aug 2020

PONE-D-20-14246R1 

Freshwater gobies 30 million years ago: new insights into character evolution and phylogenetic relationships of †Pirskeniidae (Gobioidei, Teleostei) 

Dear Dr. Reichenbacher:

I'm pleased to inform you that your manuscript has been deemed suitable for publication in PLOS ONE. Congratulations! Your manuscript is now with our production department. 

Kind regards, 

on behalf of

Dr. Giorgio Carnevale 

Academic Editor

PLOS ONE